# Beyond Tikhonov: Faster Learning with Self-Concordant Losses via Iterative Regularization

**Gaspard Beugnot**
Inria*
gaspard.beugnot@inria.fr

**Julien Mairal**
Inria†
julien.mairal@inria.fr

**Alessandro Rudi**
Inria*
alessandro.rudi@inria.fr

## Abstract

The theory of spectral filtering is a remarkable tool to understand the statistical properties of learning with kernels. For least squares, it allows to derive various regularization schemes that yield faster convergence rates of the excess risk than with Tikhonov regularization. This is typically achieved by leveraging classical assumptions called source and capacity conditions, which characterize the difficulty of the learning task. In order to understand estimators derived from other loss functions, Marteau-Ferey et al. [1] have extended the theory of Tikhonov regularization to generalized self concordant loss functions (GSC), which contain, *e.g.*, the logistic loss. In this paper, we go a step further and show that fast and optimal rates can be achieved for GSC by using the iterated Tikhonov regularization scheme, which is intrinsically related to the proximal point method in optimization, and overcomes the limitation of the classical Tikhonov regularization.

## 1   Introduction

We consider the problem of supervised learning where we want to find a prediction function $\theta$ mapping an input point $x$ living in a set $\mathcal{X}$ to a label $y$ in $\mathcal{Y}$. In this paper, we assume that $\theta$ lives in a separable Hilbert space $\mathcal{H}$ and is learned from a set of observations $(x_i, y_i)_{i=1,...,n}$ that are i.i.d. samples drawn from an unknown probability distribution $\rho$ on $\mathcal{X} \times \mathcal{Y}$. The goal is to find $\theta$ that minimizes the expected risk $L$, which is defined below along with the empirical risk $\hat{L}$:

$$L(\theta) = \int_{\mathcal{X} \times \mathcal{Y}} \ell(y, \theta(x)) \mathrm{d}\rho(x, y), \qquad \widehat{L}(\theta) = \frac{1}{n} \sum_{i=1}^{n} \ell(y_i, \theta(x_i)), \tag{1}$$

where $\ell$ is a suitable loss function comparing true labels with predictions. This paper aims for upper bounds on the excess risk for a specific estimator $\widehat{\theta}$. That is, we assume that the minimum of the expected risk is attained for some $\theta^\star$ in $\mathcal{H}$, and we want to derive *probabilistic upper bounds on the excess risk*:

$$\mathbb{P}\left[ L(\widehat{\theta}) - L(\theta^\star) > C_1 n^{-\gamma} \log \tfrac{2}{\delta} \right] \le \delta, \tag{2}$$

given some value $\delta$ in $(0, 1)$, where $C_1$ is a positive constant, and $\widehat{\theta}$ is an estimator built from the $n$ observations. The quantity $O(n^{-\gamma})$ denotes the rate of convergence of the estimator $\hat{\theta}$. A classical "slow" rate with $\gamma = 1/2$ is typically achieved by many estimators and is in fact optimal if only mild assumptions are made about the data distribution $\rho$. Even though optimal, this rate is nevertheless a worst case and faster rates with $\gamma > 1/2$ can be achieved both in theory and in practice, by making additional assumptions about the difficulty of the learning task. Originally introduced in the

---

*Inria, École normale supérieure, CNRS, PSL Research University, 75005 Paris, France
†Inria, Univ. Grenoble Alpes, CNRS, Grenoble INP, LJK, 38000 Grenoble, France

35th Conference on Neural Information Processing Systems (NeurIPS 2021).

literature of inverse problems, the so-called *source* and *capacity* conditions have been shown to be appropriate for this purpose, leading to statistical analysis with fast rates of convergence [1, 2, 3]. The optimality of results of the form (2) is characterized by comparing them with lower bounds that are available for various sets of data distributions $\rho$ [3]. Matching upper bounds with lower bounds ensures that the estimator $\widehat{\theta}$ is *optimal*, in the sense that no information is lost in the process of exploiting the data samples to compute $\widehat{\theta}$, for the given set of distributions.

In this search for optimal estimators, most of the attention has been devoted to minimizers of some function of the empirical risk $\widehat{L}$, which is defined in (Eq. (1)). Then, the key challenge is to *regularize* $\widehat{L}$ in order to achieve better generalization properties. The most widely used scheme is probably Tikhonov regularization; other examples when $\mathcal{H}$ is a RKHS include truncated regression [4], or early stopping in gradient descent algorithms [5, 6]. When the loss $\ell$ is set to least squares, it can be shown that minimizing the excess risk amounts to solving an ill-posed inverse problem [7], which led to the remarkable theory of *spectral filtering*. A large class of regularization schemes can indeed be seen as a filtering process applied to the training labels $y_i$ after regularizing the spectrum of the kernel matrix [2, 8]. Interestingly, this theory has highlighted the fact that not all regularization schemes are equal: some of them obtain fast learning rates in (2) on "easy" problem (a thorough definition is given in Section 2) while others cannot leverage this additional regularity to improve the learning rate.

Such a general analysis for least squares is made possible by the fact that a closed-form expression of the estimator is available. When considering different loss function $\ell$, the estimator $\widehat{\theta}$ is unfortunately only implicitly available as the solution of an optimization problem involving $\widehat{L}$. A step to extend least squares results to more general loss functions has been achieved by Marteau-Ferey et al. [1], who provide bounds on the form (2) for Tikhonov estimator on generalized self concordant (GSC) functions. GSC functions are three-times-differentiable functions whose third derivative is bounded by the second-derivative. In practice, they were introduced to conduct a general analysis of the Newton method in optimization [9, 10], and adapted in [11] to encompass a larger class of loss function. It includes notably the logistic regression loss, which is widely used for classification.

While Tikhonov yields fast rates of convergence in several data regimes, it is known to be unable to adapt to the whole range of learning task difficulties. More precisely, it suffers from a "saturation" effect [2], meaning that when the learning task becomes simpler, the learning rate stops improving and is suboptimal. Our paper addresses this limitation for GSC functions by considering instead the iterated Tikhonov regularization (IT) scheme. In the context of least squares, this approach consists of successively fitting the residuals. For more general loss functions, it is equivalent to performing a few steps of the proximal point method in optimization [12]. Our main result is a probabilistic upper bound on the excess risk, which is optimal given usual source and capacity conditions assumptions on the learning task, thus addressing the limitations of the classical Tikhonov regularization.

## 2 Background and Preliminaries

### 2.1 Definitions: Estimator and Loss Function

Let $\mathcal{X}$ be a Borel input space, $\mathcal{Y}$ be a vector-valued output spaces, and $\rho$ a probability distribution on $\mathcal{X} \times \mathcal{Y}$. We consider $\mathcal{H}$ to be a separable Hilbert space of functions from $\mathcal{X}$ to $\mathcal{Y}$. Given a loss function $\ell : \mathcal{Y} \times \mathcal{Y} \to \mathbb{R}$, we aim at minimizing the expected loss, while we only have access to the empirical loss – both are defined in Eq. (1). Our work provides an upper bound on the excess risk of the iterated Tikhonov estimator. For the basic case of least squares with $\mathcal{Y} = \mathbb{R}$, it is usually defined as a procedure that refits the residuals, see, e.g., §5.4 in [2]. Starting with $\widehat{\theta}_\lambda^0 = 0$, it consists of the sequence

$$\widehat{\theta}_\lambda^t = \widehat{\theta}_\lambda^{t-1} + \operatorname*{arg\,min}_{\theta \in \mathcal{H}} \left\{ \frac{1}{n} \sum_{i=1}^n \frac{1}{2} \left( y_i - \widehat{\theta}_\lambda^{t-1}(x_i) - \theta(x_i) \right)^2 + \frac{\lambda}{2} \|\theta\|^2 \right\}. \tag{3}$$

To extend this regularization to other loss function, we make the change of variable $\theta' = \widehat{\theta}_\lambda^{t-1} + \theta$ in the equation above, which yields the proximal point algorithm [12].

**Definition 1 (Iterated Tikhonov estimator a.k.a. proximal point algorithm).** *We define the iterated Tikhonov estimator with the following sequence. Given $\lambda > 0$ and $\widehat{\theta}_\lambda^0 = 0$,*

$$\widehat{\theta}_\lambda^{t+1} = \mathsf{prox}_{\widehat{L}/\lambda}(\widehat{\theta}_\lambda^t) \stackrel{\mathrm{def.}}{=} \arg\min_{\theta \in \mathcal{H}} \left\{ \widehat{L}(\theta) + \frac{\lambda}{2} \left\| \theta - \widehat{\theta}_\lambda^t \right\|^2 \right\}, \tag{4}$$

*where* $\mathsf{prox}_{\widehat{L}/\lambda}$ *denotes the proximal operator of the empirical risk* $\widehat{L}$ *rescaled by* $1/\lambda$.

*Remark* 1. In practice, the proximal operator is only computed approximately by using an optimization algorithm. Nevertheless, the benefits in terms of statistical accuracy of the iterated Tikhonov scheme are robust to inexact solutions, as long as the accuracy for solving the sub-problems (Eq. (4)) is high enough. We discuss this point in Section 3.2.

*Remark* 2. It is easy to show that the sequence of the proximal point algorithm always converges to a minimizer of the unregularized empirical risk, which is of course not what we are interested in. Instead, we consider and analyze the procedure with a fixed small number of steps $t$ and show later that optimal learning rates can be obtained by choosing an appropriate parameter $\lambda$.

*Remark* 3. When the loss is a function of a residual $y - \theta(x)$—assuming $\mathcal{Y}$ to be a vector space—as in the least square case, we recover the classical definition consisting of refitting the residual, and with $t = 1$, we recover Tikhonov.

Interestingly, our definition makes the estimator compatible with other loss functions, such as the logistic loss. More precisely, the main assumption we make on the loss is to be *generalized self concordant*. We follow the definition of [1], which is a special case of 2-self concordance introduced in [13]:

**Definition 2 (Generalized self-concordance).** *For any $z = x, y \in \mathcal{X} \times \mathcal{Y}$, the function $\ell_z : \mathcal{H} \to \mathbb{R}$ defined as $\ell_z(\theta) = \ell(y, \theta(x))$ is convex and three times differentiable. Besides, there exists a set $\phi(z) \subseteq \mathcal{H}$ s.t:*

$$\forall \theta, h, k \in \mathcal{H}, \quad \left| \nabla^3 \ell_z(\theta) [h, k, k] \right| \leq \sup_{g \in \phi(z)} |k \cdot g| \, \nabla^2 \ell_z(\theta) [k, k]. \tag{5}$$

The brackets indicate that the vectors $h, k$ and $k$ are applied to the 3-dimensional tensor $\nabla^3 \ell_z(\theta)$. The definition seems technical at first sight, but intuitively, this assumption allows to upper bound the deviation between the objective function and its local quadratic approximation. This enables a simple analysis of the Newton method for optimization, making it easy to quantify the basin of quadratic convergence [14]. On top of this, it has the benefit of encompassing a large class of loss functions, such as the logistic loss: see Example 1 in [1] for values of $\phi(z)$ with usual losses. We provide some intuition on GSC loss functions in Remark 6 in Appendix C.1.

In order to ensure the existence of the loss and its derivatives everywhere, we also need the following technical assumptions also introduced in [1], which are reasonable in practice. This ensures that both $L$ and $\widehat{L}$ are generalized self concordant too.

**Assumption 1 (Technical assumptions).** *There exists $R$ s.t $\sup_{g \in \phi(z)} \|g\| \leq R$ almost surely for $z$ drawn from the distribution $\rho$ and $|\ell_z(0)|, \|\nabla \ell_z(0)\|, \mathrm{Tr}\, \nabla^2 \ell_z(0)$ are almost surely bounded.*

The following assumption is usual in excess risk analysis [1, 15]. In our proof strategy, all the quantities are vectors and operators in $\mathcal{H}$, which makes the analysis simpler. Weakening this assumption (*e.g.* assuming that $\theta^\star \in \mathcal{L}_2(\mathcal{X})$) would require finding an equivalent of the covariance operator for GSC loss function, which constitute an interesting future direction.

**Assumption 2 (Existence of a minimizer).** *There exists $\theta^\star$ in $\mathcal{H}$ s.t $L(\theta^\star) = \inf_{\theta \in \mathcal{H}} L(\theta)$.*

Finally, following [1] we also define the *expected Hessian* and the *regularized expected Hessian* as

$$\forall \theta \in \mathcal{H}, \lambda > 0, \quad \mathbf{H}(\theta) = \mathbb{E}_{z \sim \rho} \left[ \nabla^2 \ell_z(\theta) \right], \quad \mathbf{H}_\lambda(\theta) = \mathbf{H}(\theta) + \lambda \mathbf{I},$$

and we introduce the degrees of freedom, also known as the effective dimension of the problem:

**Definition 3 (Degrees of freedom).** *The degrees of freedom is defined as:*

$$\forall \lambda, \quad \mathsf{df}_\lambda = \mathbb{E}_{z \sim \rho} \left[ \|\nabla \ell_z(\theta^\star)\|_{\mathbf{H}_\lambda^{-1}(\theta^\star)}^2 \right].$$

*where we denote by $\|\theta\|_A = \left\| A^{1/2} \theta \right\|$, with $\theta \in \mathcal{H}$, the norm induced by a positive definite operator $A$ on $\mathcal{H}$.*

*Remark* 4. The intuition about this definition is not straightforward. To better understand why this quantity is a key to characterize the amount of regularization in a learning problem, it is useful to consider the specific case of the square loss with kernels. In such a case, $\mathcal{H}$ is a reproducing kernel Hilbert space (RKHS) and $\theta(x) = \theta^\top \Phi(x)$, where $\Phi : \mathcal{X} \to \mathcal{H}$ is the kernel mapping. Then, the Hessian is constant everywhere and equal to the *covariance operator* $T = \mathbb{E}_{x \sim \rho_x}[\Phi(x) \otimes \Phi(x)]$ where $\rho_x$ is the marginal of $\rho$. Consequently, the degrees of freedom (also known as *effective dimension*) is a spectral function of $T$ which may be written as $\mathsf{df}_\lambda = \operatorname{Tr} T T_\lambda^{-1}$. This is the classical quantity which appears on the bias/variance decomposition of the excess risk, with a variance part decaying in $\operatorname{Tr} T T_\lambda^{-1}/n$, see [3].

## 2.2 Source and Capacity Conditions

We now introduce the hypotheses we make on the learning task, which will allow us to derive fast rates of convergence. They measure the difficulty of the problem and are classical in the context of learning with kernels, see *e.g.* [8, 16, 17]. It is indeed established that given an algorithm which outputs an estimator $\widehat{\theta}$, one can find a probability measure $\rho$ s.t the learning rate of the estimator is arbitrarily low, a result known as the "no-free lunch theorem" [18]. Inspired by the literature of inverse problems, two assumptions were introduced to restrict the space of considered distributions.

**Assumption 3 (Source condition).** *There exists $r > 0$ and $v$ in $\mathcal{H}$ s. t: $\theta^\star = \mathbf{H}^r(\theta^\star)v$.*

$A \mapsto A^r$ is the usual power for positive definite operators. The source condition should be seen as a smoothness assumption on $\theta^\star$, and for least square, we recover the usual definition of the source condition, that is $\theta^\star = T^r v$, with $T$ the covariance operator we previously defined. Bigger $r$ implies that the optimum can be well approximated by a few eigenvectors. Assuming $r = 0$ simplifies to $\theta^\star \in \mathcal{H}$.

The second assumption characterizes the ill-posedness of the problem:

**Assumption 4 (Capacity condition).** *There exists $\alpha > 1$, $\mathsf{s}, \mathsf{S} > 0$ s.t $\mathsf{s}\lambda^{-1/\alpha} \le \mathsf{df}_\lambda \le \mathsf{S}\lambda^{-1/\alpha}$.*

Again, for the square loss, it turns to a bound on the eigenvalue decay of the covariance operator. If $\sigma_j, e_j$ is an eigenbasis of $T$, then $\sigma_j = O(j^{-\alpha})$. Said differently, the bigger $\alpha$, the fewer directions are needed to approximate well a sample $x \sim \rho_x$ in expectation, and the easier is the learning task. This is an assumption on the input space $\mathcal{X}$ and does not imply anything on the labels $\mathcal{Y}$.

## 2.3 Previous Results

Our main result considers iterated Tikhonov *with* GSC loss functions. While iterated Tikhonov has been previously analyzed for squared loss by leveraging the theory of spectral filtering (see below), extensions to other loss functions raise several difficulties, which will be detailed in Section 3.

**Spectral filters and least squares.** As we mentioned earlier, the key insight on regularization with the square loss is that a closed-form expression of the estimator is available. By using the same notation as in Remark 4, the kernel ridge regression estimator can be for instance written

$$\widehat{\theta}_\lambda = \sum_{i=1}^n \beta_i \Phi(x_i) \quad \text{with} \quad \beta = \frac{1}{n} g_\lambda \left(\frac{K}{n}\right) y, \tag{6}$$

where $K$ is the $n \times n$ kernel matrix, $y = (y_i)_{1 \le i \le n}$ is the vector of training labels and $g_\lambda(K/n) = (K/n + \lambda I)^{-1}$. Note that $g_\lambda$ is a function acting on the spectrum of $K$, which makes it a special case of regularization by *spectral filtering*, which may be analyzed for more general functions $g_\lambda$. In particular, a key quantity for understanding the regularization effect of a filter $g_\lambda$ is the so-called *qualification*. Following [2, 8], this quantity is defined below.

**Definition 4 (Qualification of a spectral filter).** *For any $\lambda > 0$, define $g_\lambda : [0, 1] \to \mathbb{R}$ a filter function. Its qualification is the highest $q$ such that*

$$\forall \nu \le q, \quad \sup_\sigma |1 - \sigma g_\lambda(\sigma)| \sigma^\nu \le \omega_\nu \lambda^\nu, \tag{7}$$

*with $\omega_\nu$ a constant independant of $\lambda$.*

Under the source and capacity conditions, it is possible to show that the resulting estimator would enjoy an optimal rate in $n^{-\frac{\alpha(1+2r)}{1+\alpha(1+2r)}}$ if $r + 1/2 \leq q$ (where $r$ comes from the source condition). When $r + 1/2 > q$, the rate is instead of order $n^{-\frac{\alpha(1+2q)}{1+\alpha(1+2q)}}$, which is suboptimal, see *e.g.* Thm. 3.4 [17] (set the parameter $s$ to $1/2$). This illustrates the *saturation effect* of some regularization schemes. For example, Tikhonov regularization amounts to filtering with $g_\lambda : \sigma \mapsto (\sigma + \lambda)^{-1}$ and has *qualification* 1, so the parameter $r$ *saturates* at $r = 1/2$. Thus, even if $r \gg 1/2$, the excess risk of $\widehat{\theta}_\lambda$ will decay in $n^{-\frac{\alpha}{1+\alpha}}$, which is suboptimal. Designing estimators with high qualification is key to obtaining fast rates that can adapt to both hard and easy learning tasks.

**Iterated Tikhonov with the Square Loss.** We can compute the spectral filter function $g_\lambda^t$ corresponding to $t$ iterations of IT, which yields

$$g_\lambda^t : \sigma \mapsto (\sigma + \lambda)^{-1} \sum_{i=0}^{t-1} \left( \frac{\lambda}{\sigma + \lambda} \right)^i = \sigma^{-1} \left( 1 - \left( \frac{\lambda}{\sigma + \lambda} \right)^t \right). \tag{8}$$

Choosing a fixed $t$ and computing the supremum of $\sigma \mapsto |1 - \sigma g_\lambda(\sigma)| \sigma^\nu$, we find that IT estimator has qualification $t$, which is thus better than Tikhonov. IT has been thoroughly studied in the community of inverse problems, dating back to the work of [19]. It was naturally transferred to learning with kernels thanks to the aforementioned connection with inverse problems.

The link we make with the proximal point algorithm has never been studied from a statistical perspective, to the best of our knowledge, even though it has attracted a lot of attention in the optimization literature, notably with accelerated algorithms [20, 21], or variants of the proximal operator on a class of self-concordant loss functions [22]. More attention was devoted to *boosting*, where the penalty $\lambda$ is fixed but the number of iterations $t$ may go to infinity, necessitating an appropriate stopping rule [23]. Nevertheless, such a work focuses on the least square loss, where the theory of spectral filter can be applied. Finally, the proximal sequence in Eq. (4) can be cast as a constrained optimization problem related to sequential greedy approximation [24].

**Tikhonov and Generalized Self Concordant losses.** Extending the results obtained with the square loss to more general losses is challenging since there is no closed form available for the resulting estimator, and the theory of spectral filtering does not apply. Nevertheless, the case of Tikhonov regularization for GSC loss functions was treated in [1]. It is shown that the resulting estimator enjoys optimal rate as long as $r \leq 1/2$, meaning that the saturation of Tikhonov regularization is recovered in those settings. We will extend these results to the IT regularization, showing that an improved qualification can be achieved, leading to fast rates for a larger class of learning tasks.

## 3    Main Result

Our main result establishes an optimal non-asymptotic bias variance decomposition of the excess risk. It is optimal in the sense that choosing an appropriate regularization parameter $\lambda$ enables to achieve the optimal lower rates of convergence established for least squares.

**Theorem 1 (Optimal rates of IT estimator).** *Let $\delta \in (0, 1]$, and set $\lambda \in (0, \mathsf{L}_0)$, $n \geq \mathsf{N}$. The following bound on the excess risk holds with probability greater than $1 - 2\delta$:*

$$L(\widehat{\theta}_\lambda^t) - L(\theta^\star) \leq \mathsf{C}_{\mathrm{bias}} \lambda^{2s} + \mathsf{C}_{\mathrm{var}} \frac{\mathsf{df}_\lambda}{n}, \ with \ s = \min \{r + 1/2, t\}. \tag{9}$$

*If we further assume that the capacity condition holds and that the estimator does not saturate, that is $t \geq r + 1/2$, then setting*

$$\lambda = \mathsf{C}_{\mathrm{risk}} \ n^{-\frac{\alpha}{1+\alpha(2r+1)}}, \tag{10}$$

*makes the following holds with probability greater than $1 - 2\delta$:*

$$L(\widehat{\theta}_\lambda^t) - L(\theta^\star) \leq 2\mathsf{C}_{\mathrm{risk}} \ n^{-\frac{\alpha(2r+1)}{1+\alpha(2r+1)}}. \tag{11}$$

*The constants $\mathsf{L}_0, \mathsf{N}, \mathsf{C}_{\mathrm{bias}}, \mathsf{C}_{\mathrm{var}}, \mathsf{C}_{\mathrm{risk}}$ are detailed in Theorem 4 in the appendix; they are explicit and depend only on $r, \alpha, \mathsf{S}, R, t, \delta$ and the distribution $\rho$.*

**Optimal rates.** First, we note that the decay rate of the excess risk is optimal provided $t \geq r+1/2$. It means that, up to constant factors, no estimators trained on $n$ observations can benefit from a better learning rate (in the worse case sense) with the prior considered on $\rho$, that is source and capacity conditions of parameters $r, \alpha$. This leads to the second point: we see that IT has qualification $q = t$. When $t = 1$, this is Tikhonov estimator and we recover the result of [1]. This qualification shows in the bound on the bias: if $r \leq t - 1/2$, the bias is optimal in $\lambda^{2r+1}$; otherwise, it is suboptimal and decays only in $\lambda^{2t}$, which leads to higher excess risk, hence generalization error.

**Influence of $t$.** The leading multiplicative constant of the rate $\mathsf{C}_{\text{var}}$ in Eq. (9) depends linearly on the number of steps $t$, as shown in Eq. (55) in Appendix B.5. Thus, the rate in Eq. (11) is optimal in $n$ when $t = O(r)$. Letting $t$ go to infinity amounts to minimizing the empirical risk, which yield the unregularized estimator: this agrees with our bound on the excess risk, as the constant $\mathsf{C}_{\text{var}}$ would go to infinity in that case.

**Source and capacity condition.** The source and capacity conditions enable precise bounds on the bias and the variance, respectively. If they do not hold, the bias can only be bounded by $O(\lambda)$, while we can upper bound the degrees of freedom with $O(1/\lambda)$, leading to slow learning rates. If the source condition holds but the capacity condition does not, we then obtain learning rates in $n^{-2s/(2s+1)}, s = \min\{r + 1/2, t\}$, which are also optimal in these settings.

*Example: a very easy learning task.* Suppose the source condition satisfies $r = 10$ and that the capacity condition does not hold. Then, using Tikhonov estimator [1] amounts to setting $t = 1$. The generalization error would then decay as $n^{-2/3}$. On the other hand, using Iterated Tikhonov estimator with $t = 10$ would make the generalization error decay in $n^{-20/21}$, which is much better.

### 3.1 Sketch of the proof

The proof, which is fully detailed in the appendix, has the following outline:

- First, we give technical results on generalized self concordant functions;
- Then, we define the intermediate quantity in our bias-variance decomposition;
- Finally, we proceed to bounding the bias and the variance separately, which plugged together give our bound on the excess risk.

To prove the theorem above we build upon the tools from [1] on generalized self concordant functions. The resulting proof covers and simplifies the case of Tikhonov regularization (one step of iterated Tikhonov) and generalizes the rates to $r > 1/2$. We provide also a fine control of the constants, that takes into account the sequential nature of the IT estimator.

**Properties of generalized self concordant loss functions** Here, we report key properties of GSC loss functions, which are covered in depth in Appendix A. GSC loss functions are convenient to study as they come with a set of bounds on the Hessian, the gradients and the function values. Intuitively, by integrating multiple times the relation between the third and second derivative in the definition from Eq. (5), one can obtain bounds on function values. To introduce them, we first define the following function:

$$\forall \theta \in \mathcal{H}, \quad \mathsf{t}(\theta) = \sup_{z \in \mathsf{Supp}\,\rho} \sup_{g \in \phi(z)} |g \cdot \theta|. \tag{12}$$

By integrating three times the bound of the definition, one can show that:

$$L(\widehat{\theta}_\lambda^t) - L(\theta^\star) \leq \Psi\left(\mathsf{t}(\widehat{\theta}_\lambda^t - \theta^\star)\right) \left\|\widehat{\theta}_\lambda^t - \theta^\star\right\|_{\mathbf{H}(\theta^\star)}^2, \quad \Psi : t \mapsto (e^t - t - 1)/t^2. \tag{13}$$

This type of bound first appeared in [11] and was given in this form in [1]. We report it in Proposition 3 in the appendix. For instance, when $\ell$ is the square loss, $\mathsf{t} = 0$ everywhere and the r.h.s turns to $1/2\|\widehat{\theta}_\lambda^t - \theta^\star\|_T^2$, see [17, 25]. On top of this, we generalize a lower bound on the gradient:

**Lemma 1 (Stacking operator on gradient bounds).** *Let $\theta, \nu, \xi \in \mathcal{H}$, $\lambda > 0$. If $A : \mathcal{H} \to \mathcal{H}$ commutes with $\mathbf{H}(\xi)$, the following holds:*

$$e^{-\mathsf{t}(\theta-\xi)}\underline{\phi}(\mathsf{t}(\nu - \theta)) \|A(\nu - \theta)\|_{\mathbf{H}_\lambda(\xi)} \leq \|A(\nabla L_\lambda(\nu) - \nabla L_\lambda(\theta))\|_{\mathbf{H}_\lambda^{-1}(\xi)}, \tag{14}$$

*where $\underline{\phi} : t \mapsto (1 - e^{-t})/t$.*

Together with Eq. (13), this result is the workhorse of our proof for the upper bound on the excess risk. It is detailed and proven in Appendix D.

**Bias-variance decomposition.** Thanks to Eq. (13), we can relate the excess risk with the distance between estimates. This is why bounding the excess risk amounts to finding a good bias-variance decomposition. Most of the proof we find for the square loss rely on the quantity

$$\vartheta_\lambda^t = g_\lambda^t(\hat{T})\hat{T}\theta^\star, \tag{15}$$

with $\hat{T} = 1/n \sum_i \Phi(x_i) \otimes \Phi(x_i)$ the empirical covariance operator, obtained by replacing $\rho$ with the empirical distribution in Remark 4. This is basically the estimator trained on *noiseless empirical data* (*i.e.* using $\theta^\star(x_i)$ instead of $y_i$) [17, 26, 23]. Unfortunately, working with GSC function makes the spectral filtering point of view inapplicable. We need to translate a closed-form expression of the intermediate quantity with filters into the solution of an optimization problem. In our case, we can achieve the optimal bias-variance decomposition with the following quantity:

$$\begin{aligned} \vartheta_\lambda^0 &= \theta^\star, \\ \vartheta_\lambda^{k+1} &= \mathsf{prox}_{\hat{L}/\lambda}(\vartheta_\lambda^k), \quad k \geq 0. \end{aligned} \tag{16}$$

Consequently, we write

$$\|\hat{\theta}_\lambda^t - \theta^\star\|_{\mathbf{H}(\theta^\star)} \leq \|\hat{\theta}_\lambda^t - \vartheta_\lambda^t\|_{\mathbf{H}(\theta^\star)} + \|\vartheta_\lambda^t - \theta^\star\|_{\mathbf{H}(\theta^\star)}. \tag{17}$$

We recover Eq. (15) with the square loss. In [1], a different decomposition is used; we found Eq. (16) to greatly simplify the proof.

**Bounding the bias and the variance.** The first term in Eq. (17) is the *bias* of the estimator, as it goes to 0 when the regularization $\lambda$ goes to 0. By applying the lower bound on gradient values – Eq. (14) – with the definition of the proximal operator, one can express $\|\hat{\theta}_\lambda^t - \vartheta_\lambda^t\|$ function of $\|\hat{\theta}_\lambda^{t-1} - \vartheta_\lambda^{t-1}\|$. Unfolding the recursion, we obtain Theorem 2 in the appendix. It shows that the bias decreases in $O\left(\lambda^{r+1/2}\right)$ if the qualification is sufficient, *i.e.* $t \geq r + 1/2$. Otherwise, we recover the saturation experienced with least squares: the bias only decreases in $O\left(\lambda^t\right)$. Specific attention is devoted to bounding the prefactor, which is otherwise difficult to manage.

The second term in Eq. (17) is the *variance*, as it goes to 0 when the number of samples $n$ increases. Theorem 3 shows that it decays in $O(\sqrt{\mathsf{df}_\lambda/n})$. It follows closely the work of [17]. However, we cannot use the convenient fact that $\mathsf{df}_\lambda = \mathrm{Tr}\, \mathbf{H}(\theta^\star)\hat{\mathbf{H}}_\lambda^{-1}(\theta^\star)$, which is valid for least squares but not in general. Thus, we took specific care in adapting our bounds to the different regimes so as not to impact the learning rate.

Plugging these results together, we obtain the upper bound on the excess risk.

## 3.2 Optimization

The aim of this section is to extend the result of Theorem 1 to a practical case, where we only have access to an inexact solver for computing the proximal operator. Specifically, let $\epsilon > 0$ be the error (to be defined precisely in Proposition 1) made when approximating $\hat{\theta}_\lambda^t$ with $\overline{\theta}_\lambda^t$, the quantity we compute numerically. We aim for a bound of the type:

$$L(\overline{\theta}_\lambda^t) - L(\theta^\star) \leq \mathsf{C}_{\mathrm{risk}}\, n^{-\frac{\alpha(2r+1)}{1+\alpha(2r+1)}} + \epsilon.$$

The first term in the right hand side is the *statistical error*, and is optimal following the discussion of Theorem 1. The second term is the *optimization error*, which is the price to pay for approximating $\hat{\theta}_\lambda^k$ by $\overline{\theta}_\lambda^k$ with tolerance $\epsilon$. The goal is to give a simple optimization rule on the sub-problems to ensure that $\varepsilon$ is of the same order as the upper-bound for the noiseless case.

Assuming that we cannot compute the proximal operator in Eq. (4) exactly, we need to evaluate how the error in approximating $\hat{\theta}_\lambda^1$ propagates to the evaluation of $\hat{\theta}_\lambda^2$, and so on. As generalized self-concordant functions are well suited to (approximate) second-order optimization scheme, we

assume we use a solver with guarantees on a quantity called *Newton decrement*, such as the one developed in [14]. Starting from $\widehat{\theta}_\lambda^0 = \overline{\theta}_\lambda^0 = 0$, define the following for $k > 0$:

$$\widehat{\theta}_\lambda^k = \arg\min_{\theta \in \mathcal{H}} \widehat{L}_\lambda^{k-1}(\theta) \stackrel{\text{def.}}{=} \widehat{L}(\theta) + \frac{\lambda}{2} \left\| \theta - \widehat{\theta}_\lambda^{k-1} \right\|^2, \quad \nu_\lambda^k(\theta) \stackrel{\text{def.}}{=} \left\| \nabla \widehat{L}_\lambda^{k-1}(\theta) \right\|_{\widehat{\mathbf{H}}_\lambda^{-1}(\theta)}, \quad (18)$$

$$\overline{\theta}_\lambda^k \approx \arg\min_{\theta \in \mathcal{H}} \overline{L}_\lambda^{k-1}(\theta) \stackrel{\text{def.}}{=} \widehat{L}(\theta) + \frac{\lambda}{2} \left\| \theta - \overline{\theta}_\lambda^{k-1} \right\|^2, \quad \overline{\nu}_\lambda^k(\theta) \stackrel{\text{def.}}{=} \left\| \nabla \overline{L}_\lambda^{k-1}(\theta) \right\|_{\widehat{\mathbf{H}}_\lambda^{-1}(\theta)}. \quad (19)$$

$\overline{\theta}_\lambda^k$ approximates the proximal operator evaluated on $\overline{\theta}_\lambda^{k-1}$, and $\overline{L}_\lambda^{t-1}$ is the function we manipulate at step $t$. If the optimization was carried without error in Eq. (19), we would have $\overline{L}^{t-1} = \widehat{L}^{t-1}$. The quality of the approximation is measured with the Newton decrement of Eq. (19), see *e.g*, Lemma 6 of [14]. We need to enforce a bound on the true Newton decrement in Eq. (18) when we only have access to $\overline{L}_\lambda^{t-1}$. The next proposition gives a simple rule to achieve this.

**Proposition 1 (Error propagation with proximal sequence).** *Let $\epsilon > 0$ the target precision. Assume that we can solve each sub-problem with precision $\bar{\epsilon}_k$:*

$$\forall k \in \{1, \ldots, t\}, \quad \overline{\nu}_\lambda^{k-1}\left(\overline{\theta}_\lambda^k\right) \le \bar{\epsilon}_k = \epsilon \frac{1.4^{k-t}}{t},$$

*and that $\epsilon \le \sqrt{\lambda}/(2R)$. This suffice to achieve an error $\epsilon$ on the target function:*

$$\nu_\lambda^{t-1}\left(\overline{\theta}_\lambda^t\right) \le \epsilon.$$

This is a specialized version of Proposition 7, whose proof is detailed in the appendix. Intuitively, this means that enforcing a geometrically higher precision on the first steps is sufficient to obtain high precision on the final estimate. To compute IT's estimator in practice, one would need to solve $t$ optimization problem with decreasing precision. As second order schemes have double logarithmic complexity w.r.t the precision $\epsilon$, the complexity of computing the proximal sequence of IT with tolerance $\epsilon$ would be only (up to logarithm term) $t$ times bigger than estimating Tikhonov estimator with tolerance $\epsilon$. In practice, when learning with kernels, one would use the representer theorem and aim at estimating $\beta$ in $\mathbb{R}^n$ as in Eq. (6) [27]. This results in an optimization problem with $n$ observations in dimension $n$, with complexity $O(n^3)$. A practical implementation could use Nyström projection to avoid this cubic computational burden in the number of samples. The statistical effects of such projection are well studied with Tikhonov regularization [14, 15]; their effect on other regularization scheme is an interesting future research direction.

This proposition can be used directly to bound the excess risk with inexact solvers.

**Proposition 2 (Upper bound on the excess risk with inexact solvers).** *Let $\delta \in (0, 1)$ and assume that the* statistical *assumptions of Theorem 1 hold as well as the* optimization *assumptions of Proposition 1. Then, the following bound on the excess risk holds with probability greater than $1 - 2\delta$:*

$$L(\overline{\theta}_\lambda^t) - L(\theta^\star) \le 2\mathsf{C}_{\text{risk}} \; n^{-\frac{\alpha(2r+1)}{1+\alpha(2r+1)}} + \mathsf{E}_{1/2}\epsilon, \quad s = \min\{r + 1/2, t\} \quad (20)$$

*with $\mathsf{C}_{\text{risk}}$ as in Theorem 1 and $\mathsf{E}_{1/2} \le 4.3 \cdot 10^3$.*

This is a specialized version of Proposition 8 proved in the appendix. The first term is the statistical excess risk, whereas the second term in $\epsilon$ is the price we pay for inexact approximation. For the sake of clarity, crude upper bounds were used (notably $\widehat{\mathbf{H}}_\lambda^{-1/2}(\cdot) \le \mathsf{B}_2^\star/\sqrt{\lambda}$) at the expanse of big constants. They can be expected to be an order of magnitude lower in practice.

**Setting $t$ in real application.** In classical machine learning settings, we do not have access to the source condition parameter $r$. The number of proximal steps $t$ can be seen as an hyperparameter, which is chosen by cross-validation. One would run the algorithm and test the resulting error on a validation set for each iteration, and keep doing proximal steps as long as the validation loss improves.

## 4 Experiments

The purpose of the experiments is to illustrate the saturation effect of the Tikhonov estimator when $r \gg 1/2$, and see how the saturation is overcome by iterated Tikhonov IT. We also show that the statistical rates we derive are achieved both in theory and in practice on synthetic data with well-controlled source and capacity conditions.

**Settings.** To that end, we use a synthetic binary classification data set for which we know the source and capacity condition parameters $r$ and $\alpha$ by design. Then, we study the performance of $\mathsf{IT}(t)$, $t \in \{1, \ldots, 8\}$, trained with the logistic loss, which satisfies Definition 2 about generalized self-concordant functions. Related experiments were conducted in the context of kernel ridge regression with synthetic data in [16], which we follow here. Specifically, we use splines of order $\alpha$ to define a kernel matrix:

$$K(x, z) = \Lambda_\alpha(x, z) = \sum_{k \in \mathbb{Z}} \frac{e^{2i\pi k(x-z)}}{|k|^\alpha},$$

for which a closed form expression is available as soon as $\alpha$ is a positive even integer (see for instance Eq (2.1.7) in [28]). We then use $\mathcal{X} = [0, 1]$, $\rho_x$ is the uniform distribution, and $\theta^\star(x) = \Lambda_{(r+1/2)\alpha+1/2}(0, \cdot)$, which may be shown to live in the RKHS $\mathcal{H}$ of $K$. Then, it is possible to show that the source and capacity assumption are satisfied with value $r, \alpha$, see [16].

Finally, we design the distribution $\rho_{y|x}$ of the labels such that $\theta^\star$ is indeed the minimizer of the risk over $\mathcal{H}$. This may be ensured if $\theta^\star$ coincides with the minimizer of the risk over the set of measurable functions, which has the following form under mild assumptions (see Eq. (3) in [26]):

$$\theta^\star(x) = \arg\min_z \mathbb{E}_{y|x}\left[\ell(y, z)\right]. \tag{21}$$

The previous relation can be satisfied by choosing $\rho_{y|x}$ accordingly. More precisely, we need

$$\mathcal{Y} = \{-1, 1\}, \quad \mathbb{P}(y = 1 \mid x) = \left(1 + e^{-\theta^\star(x)}\right)^{-1}, \quad \mathbb{P}(y = -1 \mid x) = \left(1 + e^{\theta^\star(x)}\right)^{-1},$$

which ensures that Eq. (21) holds – see details in Appendix E.3. To our knowledge, this is the first synthetic dataset with given source and capacity condition for classification tasks. For each $\lambda, t$, we sample $n$ points uniformly on $[0, 1]$, evaluate $\theta^\star$, the observed labels $y_i$, and $\widehat{\theta}_\lambda^t$. We evaluate the excess risk $L(\widehat{\theta}_\lambda^t) - L(\theta^\star)$ with Monte Carlo sampling. We then report the *lowest excess risk* achieved across the regularization $\lambda$, and the *optimal regularization* used to achieve this loss. We plot lines of slope $2s\alpha/1+2s\alpha$ and $\alpha/1+2s\alpha$ respectively, with $s = (r + 1/2) \wedge t$ in order to compare the statistical rates achieved in practice and in theory.

**Results.** Results for the logistic loss are available in Fig. 1 and we also present results with least squares where the noise is Gaussian in Appendix E.2. We set $\alpha = 2$, $r \in \{1/4, 41/4\}$, and we study the performance of Iterated Tikhonov estimators with $t \in \{1, 3, 8\}$. $t = 1$ corresponds to Tikhonov estimator and saturates at $r = 1/2$. $\mathsf{IT}(3)$ and $\mathsf{IT}(8)$ saturates at $r = 5/2$ and $r = 15/2$ respectively. Consequently, all estimators have optimal rates on the difficult task with $r = 1/4$; however, only IT exploits the additional regularity of the easy task, with $r = 41/4$. This experimentally shows that better sample complexity can be achieved when the learning task is easier and $t$ is high, matching the rates predicted in Theorem 4, which are $n^{-\alpha(1+2s)/1+\alpha(1+2s)}$, with $s = \min\{r, t - 1/2\}$. Learning rates were estimated with an ordinary least square regression in log-log scale, and are given in Table 1, where they are compared with the theoretical values. To conclude, we observe a slight improvement in absolute value of the excess risk in the range $r \ll t$, suggesting that IT is useful even when the learning task is hard. This could be because of lower constants for high $t$: *e.g.* we show that $\mathsf{C}_{\mathrm{bias}}$ decays in $1/t^r$ when $t \geq r + 1/2$, see Theorem 2 in the appendix. We report in the appendix additional experimental results such as plots with the chosen regularization $\lambda$ as a function of $n$, and plots on the ratio between the excess risk of $\mathsf{IT}(t)$ and Tikhonov, to show that the former is consistently better than the latter on easy tasks.

## 5 Conclusion

This paper studies a well-known regularization scheme for least square, and extend it for the first time to other loss functions, which notably contain the logistic loss used for classification. We prove

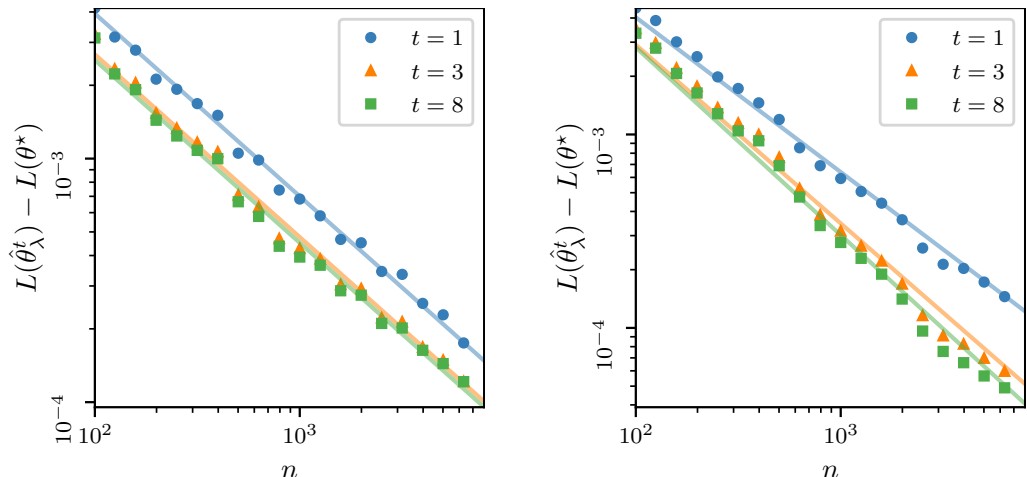

Figure 1: Excess risk for various Iterated Tikhonov estimators as a function of $n$. **Colors**: $t = 1$ (Tikhonov) estimator is shown in blue; $t = 3, 8$ in green, orange. **Left**: from a difficult problem, $r = 1/4, \alpha = 2$. **Right**: easy problem, $r = 41/4, \alpha = 2$. Plain lines are predicted by theory, with slope $-\alpha(1+2s)/1+\alpha(1+2s)$, $s = \min\{r, t - 1/2\}$ (see main text). All plots are averaged over 100 runs of the optimization procedure with different initialization.

Table 1: Learning rate coefficients for capacity condition $\alpha = 2$ and various source condition assumption $r$. We estimate $\gamma$ with ordinary least square with the model $L(\widehat{\theta}_\lambda^t) - L(\theta^\star) \propto n^{-\gamma}$. We display the coefficient we expect in theory, and the one we estimate.

|  | $r$ | 0.25 | 3.25 | 10.25 |
|---|---|---|---|---|
| $t = 1$ | Theory | 0.75 | 0.80 | 0.80 |
|  | Estimation | 0.71 | 0.73 | 0.72 |
| $t = 3$ | Theory | 0.75 | 0.92 | 0.92 |
|  | Estimation | 0.75 | 0.83 | 0.87 |
| $t = 8$ | Theory | 0.75 | 0.94 | 0.97 |
|  | Estimation | 0.79 | 0.95 | 0.98 |

that Iterated Tikhonov, corresponding to proximal point iterations, has optimal learning rates and higher qualification than Tikhonov, and as such could outperform it on easy tasks. We extend the scope of the theory of learning with generalized self concordant loss functions beyond standard Tikhonov regularization, which fills a gap in the previous theory, showing that it is possible to be fully adaptive to the regularity of the learning problem, without saturation effects. On top of this, we gave sufficient conditions to compute the estimator in practice, which is nontrivial by its sequential nature. Interesting research directions include related regularization schemes, such as boosting, but also implementations of the iterated Tikhonov procedure with sketching techniques as Nyström projections. The goal is to derive algorithms that are both optimal, in terms of statistical guarantees, and with reduced computational complexity, which is an aspect we will address in future work.

**Acknowledgments**

A.R. acknowleges support of the French government under management of Agence Nationale de la Recherche as part of the "Investissements d'avenir" program, reference ANR-19-P3IA-0001 (PRAIRIE 3IA Institute). A.R. acknowledges support of the European Research Council (grant REAL 947908). J. Mairal was supported by the ERC grant number 714381 (SOLARIS project) and by ANR 3IA MIAI@Grenoble Alpes, (ANR19-P3IA-0003).

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
