# Appendix

## Table of Contents

## A   Settings, notations and assumptions

Given a separable Hilbert space $\mathcal{H}$, $\|\cdot\|$ denotes the norm in $\mathcal{H}$. For any operator $A$ on $\mathcal{H}$, $\|A\|$ denotes its operator norm, and $\operatorname{Tr} A$ its trace norm. If $A$ is a p.d operator, we denote by $\|\cdot\|_A = \|A^{1/2}\cdot\|$ the norm induced by $A$. We denote $\|A\|_{HS}$ the Hilbert Schmidt norm of $A$. We use the short-hand notation

$$A_\lambda = A + \lambda \mathbf{I},$$

where $\mathbf{I}$ is the identity. We denote by $a \wedge b$ the minimum of $\{a, b\}$, and $a \vee b$ its maximum.

### A.1   Settings and technical assumptions

The settings in this subsection are the same as in [1]. We report them for completeness.

Let $\mathcal{X}$ a Borel input space, $\mathcal{Y}$ be a vector-valued output spaces, and $\rho$ a probability distribution on $\mathcal{X} \times \mathcal{Y}$. We consider $\mathcal{H}$ to be a separable Hilbert space of functions from $\mathcal{X}$ to $\mathcal{Y}$. We consider a loss function $\ell : \mathcal{Y} \times \mathcal{Y} \to \mathbb{R}$ for measuring the fit between predictions and true labels. Given $n$ observations $(x_1, y_1), \ldots, (x_n, y_n)$ i.i.d according to $\rho$, the goal is to build a measurable function $\widehat{\theta}$, which minimizes the expected loss

$$L(\widehat{\theta}) = \mathbb{E}_{x,y \sim \rho}\left[ \ell(y, \widehat{\theta}(x)) \right].$$

In this paper, we evaluate the quality of the estimator with probabilistic upper bounds on the *excess risk*

$$L(\widehat{\theta}) - \inf_{\theta \in \mathcal{H}} L(\theta) \leq K n^{-\gamma},$$

with probability greater than $1 - \delta$. The rate of decay $\gamma$ is referred to as the *learning rate* of the estimator. Our main assumption on the loss function is to be generalized self-concordant (GSC).

**Assumption 5 (Generalized Self-Concordance).** *For any $z = x, y \in \mathcal{X} \times \mathcal{Y}$, the function $\ell_z : \mathcal{H} \to \mathbb{R}$ defined as $\ell_z(\theta) = \ell(y, \theta(x))$ for $\theta \in \mathcal{H}$ is convex and three times differentiable. Besides, there exists a set $\phi(z) \subset \mathcal{H}$ s.t*

$$\forall \theta \in \mathcal{H}, \, \forall h, k \in \mathcal{H}, \quad \left| \nabla^3 \ell_z(\theta) [h, k, k] \right| \leq \sup_{g \in \phi(z)} |k \cdot g| \, \nabla^2 \ell_z(\theta) [k, k].$$

Next, we introduce the following quantities.

**Definition 5 (Useful quantities).** *Let $\theta \in \mathcal{H}$. The following quantities are independant of the random variable $z \sim \rho$, either by taking the supremum over the support of $\rho$ or by considering the expectation. Define:*

- *uniform bounds on the derivatives:*

$$\mathsf{B}_1(\theta) = \sup_{z \in \mathsf{Supp}\,\rho} \|\nabla \ell_z(\theta)\|, \quad \mathsf{B}_2(\theta) = \sup_{z \in \mathsf{Supp}\,\rho} \mathrm{Tr}\, \nabla^2 \ell_z(\theta);$$

- *the Hessian of the expected and empirical loss:*

$$\mathbf{H}(\theta) = \nabla^2 \mathbb{E}\left[\ell_z(\theta)\right], \quad \widehat{\mathbf{H}}(\theta) = \frac{1}{n} \sum_{i=1}^n \nabla^2 \ell_{z_i}(\theta);$$

- *the function $\mathsf{t}$, s.t:*

$$\mathsf{t}(\theta) = \sup_{z \in \mathsf{Supp}\,\rho} \sup_{g \in \phi(z)} |\theta \cdot g|.$$

We make technical assumption to ensure that the loss function and its derivatives are well defined everywhere and that we can exchange expectation and derivative.

**Assumption 6 (Technical assumptions).** *There exists $R$ s.t $\sup_{g \in \phi(z)} \|g\| \leq R$ almost surely; $|\ell_z(0)|, \|\nabla \ell_z(0)\|, \mathrm{Tr}\,\mathbf{H}(\nabla^2 \ell_z(0))$ are almost surely bounded.*

Using Prop. 2 of [1], we have that $\mathsf{B}_1(\theta), \mathsf{B}_2(\theta), L(\theta), \nabla L(\theta), \mathbf{H}(\theta)$ exist for all $\theta \in \mathcal{H}$, and

$$\nabla L(\theta) = \mathbb{E}\left[\nabla \ell_z(\theta)\right], \quad \mathbf{H}(\theta) = \mathbb{E}\left[\nabla^2 \ell_z(\theta)\right].$$

Finally, $\mathbf{H}(\theta)$ is trace-class, that is its trace is finite for any $\theta \in \mathcal{H}$. The same properties hold when considering $\hat{\rho}$ instead of $\rho$, that is for the quantities $\widehat{L}(\theta), \nabla \widehat{L}(\theta)$ and $\widehat{\mathbf{H}}(\theta)$.

We make three key assumptions to obtain our learning rate.

**Assumption 7 (Existence of a minimizer).** *There exists a minimizer of $L$ in $\mathcal{H}$. There is $\theta^\star \in \mathcal{H}$ s.t*

$$L(\theta^\star) = \inf_{\theta \in \mathcal{H}} L(\theta^\star).$$

**Assumption 8 (Source condition).** *There exists $r > 0$ and $v \in \mathcal{H}$ s. t*

$$\theta^\star = \mathbf{H}^r(\theta^\star)v.$$

The third assumption qualifies the ill-posedness of the problem:

**Assumption 9 (Capacity condition).** *There exists $\alpha > 1$, $\mathsf{s}, \mathsf{S} > 0$ s.t*

$$\mathsf{s}\lambda^{-1/\alpha} \leq \mathsf{df}_\lambda \leq \mathsf{S}\lambda^{-1/\alpha}.$$

To understand the source and capacity condition, one must pay attention to the counterpart of the covariance operator for GSC loss function, that is the expected hessian at optimality. It is denoted with $\mathbf{H}(\theta^\star)$ throughout the paper. The source and capacity conditions are assumptions on the eigendecomposition of this operator. To better quanitfy these assumptions, take $\sigma_j, e_j$ an eigenbasis of $\mathbf{H}(\theta^\star)$, with $\sigma_j > \sigma_{j+1}$.

The source condition is a smoothness assumption on $\theta^\star$. It amounts to assuming that the eigende-composition of $\theta^\star$ on the basis of the Hessian decays faster than its spectrum. Indeed, rewriting Assumption 8 we obtain

$$\|v\|^2 = \sum_{j \geq 1} \sigma_j^{-2r} \langle \theta^\star, e_j \rangle^2 < +\infty.$$

Assuming $r = 0$ simplifies to $\theta^\star \in \mathcal{H}$. Bigger $r$ implies that the optimum can be well approximated by the first few eigenvectors (as $(\sigma_j^{-2r})_j$ goes quickly to infinity).

Similarly, the capacity condition is an assumption on the decay of the spectrum of the Hessian. Specifically, it assumes that the spectrum decays polynomially, i.e $\sigma_j \sim j^{-\alpha}$. As this operator is compact, we have $\alpha > 1$ for the $\sum_j j^{-\alpha}$ to be summable. Bigger $\alpha$ gives easier input space $\mathcal{X}$.

See Section 2 in the main body of the paper for a discussion on the significance of these assumptions.

## A.2 Basic results on GSC loss functions

Here, we present Prop. 4 of [1], which we then extend with an additional lemma.

**Proposition 3 (Properties of GSC functions).** *Let $\theta, \nu \in \mathcal{H}$, $\lambda \geq 0$. The following properties hold:*

$$\widehat{\mathbf{H}}_\lambda(\theta) \preceq e^{\mathsf{t}(\theta - \nu)} \widehat{\mathbf{H}}_\lambda(\nu) \tag{22}$$

$$\left\| \nabla \widehat{L}_\lambda(\theta) - \widehat{L}_\lambda(\nu) \right\|_{\widehat{\mathbf{H}}_\lambda^{-1}(\theta)} \leq \|\theta - \nu\|_{\widehat{\mathbf{H}}_\lambda(\theta)} \overline{\phi}\left(\mathsf{t}(\theta - \nu)\right) \tag{23}$$

$$L_\lambda(\theta) - L_\lambda(\nu) - \nabla L_\lambda(\nu) \cdot (\theta - \nu) \leq \Psi\left(\mathsf{t}(\theta - \nu)\right) \|\theta - \nu\|_{\mathbf{H}_\lambda(\theta)}^2 \tag{24}$$

*where $\underline{\phi} : t \mapsto (1 - e^{-t})/t$ and $\Psi : t \mapsto (e^t - t - 1)/t^2$. Moreover, if $\nu, \xi \in \mathcal{H}$, $A : \mathcal{H} \to \mathcal{H}$ commutes with $\mathbf{H}(\xi)$, then the following holds:*

$$e^{-\mathsf{t}(\theta - \xi)} \underline{\phi}\left(\mathsf{t}(\nu - \theta)\right) \|A(\nu - \theta)\|_{\mathbf{H}_\lambda(\xi)} \leq \|A(\nabla L_\lambda(\nu) - \nabla L_\lambda(\theta))\|_{\mathbf{H}_\lambda^{-1}(\xi)} \tag{25}$$

We slightly modify the lower bound gradient, which is crucial for obtaining higher qualification with IT.

**Lemma 2 (Stacking operator on gradient bounds).** *Let $\theta, \nu, \xi \in \mathcal{H}$, $\lambda > 0$. If $A : \mathcal{H} \to \mathcal{H}$ commutes with $\mathbf{H}(\xi)$, the following holds:*

$$e^{-\mathsf{t}(\theta - \xi)} \underline{\phi}\left(\mathsf{t}(\nu - \theta)\right) \|A(\nu - \theta)\|_{\mathbf{H}_\lambda(\xi)} \leq \|A(\nabla L_\lambda(\nu) - \nabla L_\lambda(\theta))\|_{\mathbf{H}_\lambda^{-1}(\xi)}.$$

*Proof.* Defining $v_s = \theta + s(\nu - \theta)$ for $s \in \{0, 1\}$, we have:

$$A^2 \left(\nabla L_\lambda(\nu) - \nabla L_\lambda(\theta)\right) = A^2 \int_0^1 \mathbf{H}_\lambda(v_s) (\nu - \theta) \, \mathrm{d}s,$$

which implies $\quad \left\langle A^2 \left(\nabla L_\lambda(\nu) - \nabla L_\lambda(\theta)\right), \nu - \theta \right\rangle = A^2 \int_0^1 \left\langle \mathbf{H}_\lambda(v_s)(\nu - \theta), \nu - \theta \right\rangle \mathrm{d}s.$

We may then use the lower bound on the Hessian from Eq. (22),

$$\mathbf{H}_\lambda(v_s) \succeq \mathbf{H}_\lambda(\xi) e^{-\mathsf{t}(v_s - \xi)} \succeq \mathbf{H}_\lambda(\xi) e^{-\mathsf{t}(\theta - \xi)} e^{-s\mathsf{t}(\nu - \theta)},$$

where the second inequality comes from $\mathsf{t}$ satisfying the triangle inequality. Plugging this in the previous equation and using the fact that $\mathbf{H}(\xi)$ and $A$ commute, we have that:

$$\int_0^1 \left\langle A^2 \mathbf{H}_\lambda(v_s)(\nu - \theta), \nu - \theta \right\rangle \mathrm{d}s \geq e^{-\mathsf{t}(\theta - \xi)} \int_0^1 e^{-s\mathsf{t}(\nu - \theta)} \mathrm{d}s \; \left\langle \mathbf{H}_\lambda(\xi) A(\nu - \theta), A(\nu - \theta) \right\rangle$$

$$= e^{-\mathsf{t}(\theta - \xi)} \underline{\phi}\left(\mathsf{t}(\nu - \theta)\right) \left\langle \mathbf{H}_\lambda(\xi) A(\nu - \theta), A(\nu - \theta) \right\rangle,$$

which gives the lower bound

$$e^{-\mathsf{t}(\theta - \xi)} \underline{\phi}\left(\mathsf{t}(\nu - \theta)\right) \|A(\nu - \theta)\|_{\mathbf{H}_\lambda(\xi)}^2 \leq \left\langle A^2 \left(\nabla L_\lambda(\nu) - \nabla L_\lambda(\theta)\right), \nu - \theta \right\rangle. \tag{26}$$

On the other hand, with Cauchy Schwartz inequality, we obtain:

$$\left\langle A^2 (\nabla L_\lambda(\nu) - \nabla L_\lambda(\theta)), \nu - \theta \right\rangle \leq \|A(\nabla L_\lambda(\nu) - \nabla L_\lambda(\theta))\|_{\mathbf{H}_\lambda^{-1}(\xi)} \|A(\nu - \theta)\|_{\mathbf{H}_\lambda(\xi)}. \tag{27}$$

Combining the inequalities Eqs. (26) and (27) and dividing by $\|A(\nu - \theta)\|_{\mathbf{H}_\lambda(\xi)}$, we obtain the result needed. $\qquad \square$

# B Proof of Theorem 1

## B.1 Error decomposition

Thanks to Eq. (24), the excess risk is bounded by the distance between estimate in $\mathbf{H}(\theta^\star)$ norm with

$$L_\lambda(\widehat{\theta}_\lambda^t) - L_\lambda(\theta^\star) \leq \Psi\left(\mathsf{t}(\widehat{\theta}_\lambda^t - \theta^\star)\right) \left\|\widehat{\theta}_\lambda^t - \theta^\star\right\|_{\mathbf{H}_\lambda(\theta^\star)}^2.$$

In order to compute $\|\widehat{\theta}_\lambda^t - \theta^\star\|_{\mathbf{H}(\theta^\star)}$, we need to go through an intermediate quantity $\vartheta$. In the context of least squares and spectral filters, such quantity is usually defined to be

$$\vartheta = g_\lambda(\hat{T})\hat{S}^*\hat{S}\theta^\star, \tag{28}$$

where:

- $\hat{T} = \hat{S}^*\hat{S}$ is the *empirical covariance operator*, equal to $\sum_{i=1}^n \Psi(x_i) \otimes \Psi(x_i)$ when $\mathcal{H}$ is a RKHS with feature map $\Psi$ (see Remark 4);
- $\hat{S} : \mathcal{H} \to \mathbb{R}^n$ is the *sampling operator*, with $\hat{S}\theta = 1/\sqrt{n}(\theta(x_i), \ldots, \theta(x_n))$;
- Its dual is $\hat{S}^* : \mathbb{R}^n \to \mathcal{H}$, with $\hat{S}^*y = 1/\sqrt{n}\sum_{i=1}^n y_i\Phi(x_i)$;

see [17] for details. Thus, the quantity in Eq. (15) can be seen as the estimator trained on the *empirical noiseless distribution*, where we use $\hat{S}\theta^\star$ instead of $y = (y_i)_{1\leq i\leq n}$. It is optimal in the sense that its bias $\|\vartheta - \theta^\star\|_{\hat{T}}$ will be of the order of $\lambda^{r+1/2}$ and its variance $\left\|\widehat{\theta}_\lambda^t - \vartheta\right\|_{\hat{T}}$ of the order of $\mathsf{df}_\lambda/n$, leading to the optimal rates for least squares [3].

Expressing the quantity above as a proximal sequence is the key insight of the proof. It turns out that the following quantity obtains the same optimal decomposition.

**Definition 6 (Error decomposition).** *Define the following quantity:*

$$\vartheta_\lambda^0 = \theta^\star$$
$$\vartheta_\lambda^{k+1} = \mathsf{prox}_{\hat{L}/\lambda}(\vartheta_\lambda^k), \quad k \geq 0$$

*Remark* 5. In fact, the estimator above, when expressed with filters, has its (bias, variance) equals to the (variance, bias) of the estimator of Eq. (28). It is easy to change the intermediate quantity of Definition 6 to match, but it introduces unnecessary burden with the notations.

The purpose of next sections is to bound

$$\left\|\widehat{\theta}_\lambda^t - \theta^\star\right\|_{\mathbf{H}(\theta^\star)} \leq \left\|\widehat{\theta}_\lambda^t - \vartheta_\lambda^t\right\|_{\mathbf{H}(\theta^\star)} + \left\|\vartheta_\lambda^t - \theta^\star\right\|_{\mathbf{H}(\theta^\star)}. \tag{29}$$

The first term will be the *bias* of the estimator (decreases with $\lambda/t$) while the second one will be the *variance* (decreases with $t/\lambda$ and $n$). The intermediate quantity of Definition 6 being very close to the one of Eq. (28) used in [17], it is natural that the proof follows similarly.

## B.2 Bounding the bias

Here, we proceed in bounding the bias, that is the quantity $\|\widehat{\theta}_\lambda^t - \vartheta_\lambda^t\|_{\mathbf{H}(\theta^\star)}$.

**Theorem 2 (Improved qualification of Iterated Tikhonov estimator).** *Let $\delta \in (0, 1]$. Recall the source condition of parameter $r, \|v\|$. Define the following conditions on the number of samples:*

$$\mathsf{H}_1 : n \geq 24\frac{\mathsf{B}_2^\star}{\lambda}\log\frac{16\mathsf{B}_2^\star}{\lambda\delta},$$

$$\mathsf{H}_{1b} : n \geq 8\frac{\mathsf{B}_2^{\star 2}}{\lambda^2}\log^2\frac{4}{\delta},$$

$$\mathsf{H}_2 : n \geq 2\left[1 \vee \left(\frac{2\mathsf{B}_2^\star(t - 1/2)^r}{\lambda^{s-1/2}}\right)^2\right]\log\frac{4}{\delta},$$

*Now assume:*

$$\begin{aligned} \mathsf{H}_1 \quad &\text{if } r \le 1/2, \\ \mathsf{H}_1 + \mathsf{H}_{1b} \quad &\text{if } 1/2 < r \le 1, \\ \mathsf{H}_1 + \mathsf{H}_2 \quad &\text{if } r > 1. \end{aligned}$$

*Then, with probability greater than $1 - \delta$:*

$$\left\| \widehat{\theta}_\lambda^t - \vartheta_\lambda^t \right\|_{\mathbf{H}(\theta^\star)} \le \sqrt{2} \mathsf{T}(r,t) \mathsf{P}_\lambda^t \lambda^s, \tag{30}$$

*with $s = (r + 1/2) \wedge t$,*

$$\mathsf{T}(r,t) = \begin{cases} \|v\| \left(1 \vee (\mathsf{B}_2^\star + \lambda)\right) 2^r & \text{if } r \le 1, \\ \|v\| \frac{w(r)+r}{(t-1/2)^r} & \text{if } r > 1 \text{ and } r + 1/2 < t \\ \|v\| \frac{w(r)}{(t-1/2)^r} + \mathsf{B}_2^{\star\, r-t+1/2} & \text{if } r > 1 \text{ and } r + 1/2 \ge t, \end{cases} \tag{31}$$

$w(r) = r 2^{\lfloor r \rfloor + 1} \mathsf{B}_2^{\star\, r}$, *and:*

$$\mathsf{P}_\lambda^t \stackrel{\text{def.}}{=} \prod_{k=1}^{t} \underline{\phi}^{-1} \left( \hat{\mathsf{t}}(\widehat{\theta}_\lambda^k - \vartheta_\lambda^k) \right) e^{\hat{\mathsf{t}}(\vartheta_\lambda^k - \theta^\star)}.$$

This term is the optimal bias for LS with the usual excess risk decomposition. The saturation effect is explicit: we go from a bias decay in $\lambda^t$ when $t \le r + 1/2$ to $\lambda^r$ when the source condition saturates IT's regularization. That is, IT's estimator has a qualification of $t$, in the sense that it can exploits source condition up to $r = t - 1/2$. If $r > t - 1/2$, the estimator saturates and the learning rate becomes suboptimal.

*Proof.* This proof simply relies on the upper bound on gradients enabled by GSC functions. We will use Lemma 2 for that purpose. Also, we will use the definition of a proximal sequence; that is, we have that

$$\forall k \le t, \quad \nabla \widehat{L}(\widehat{\theta}_\lambda^k) + \lambda(\widehat{\theta}_\lambda^k - \widehat{\theta}_\lambda^{k-1}) = 0,$$

which is just another way of saying that we perform implicit gradient steps of size $1/\lambda$.

**Changing the norm.** We first change the norm we operate on:

$$\left\| \widehat{\theta}_\lambda^t - \vartheta_\lambda^t \right\|_{\mathbf{H}(\theta^\star)} \le \left\| \widehat{\mathbf{H}}_\lambda^{-1/2}(\theta^\star) \mathbf{H}^{1/2}(\theta^\star) \right\| \left\| \widehat{\theta}_\lambda^t - \vartheta_\lambda^t \right\|_{\widehat{\mathbf{H}}_\lambda(\theta^\star)}$$

$$\le \left\| \widehat{\mathbf{H}}_\lambda^{-1/2}(\theta^\star) \mathbf{H}_\lambda^{1/2}(\theta^\star) \right\| \left\| \widehat{\theta}_\lambda^t - \vartheta_\lambda^t \right\|_{\widehat{\mathbf{H}}_\lambda(\theta^\star)}.$$

We bound the operator norm using Proposition 9 in Appendix D, with $\mathcal{F}_\lambda = \mathsf{B}_2^\star / \lambda$. We obtain:

$$\mathsf{H}_1 : n \ge 24 \frac{\mathsf{B}_2^\star}{\lambda} \log \frac{8\mathsf{B}_2^\star}{\lambda \delta} \implies \left\| \widehat{\mathbf{H}}_\lambda^{-1/2}(\theta^\star) \mathbf{H}_\lambda^{1/2}(\theta^\star) \right\| \le \sqrt{2}. \tag{32}$$

We now proceed in bounding the distance between estimates, that is the quantity $\|\widehat{\theta}_\lambda^t - \vartheta_\lambda^t\|_{\widehat{\mathbf{H}}_\lambda(\theta^\star)}$. We denote

$$s = (r + 1/2) \wedge t. \tag{33}$$

**Upper bound on gradients.** Use Lemma 2 on $\widehat{L}_\lambda$ to have:

$$\left\| \widehat{\theta}_\lambda^t - \vartheta_\lambda^t \right\|_{\widehat{\mathbf{H}}_\lambda(\theta^\star)} \le \underline{\phi}^{-1} \left( \hat{\mathsf{t}}(\widehat{\theta}_\lambda^t - \vartheta_\lambda^t) \right) e^{\hat{\mathsf{t}}(\vartheta_\lambda^t - \theta^\star)} \left\| \nabla \widehat{L}_\lambda(\widehat{\theta}_\lambda^t) - \nabla \widehat{L}_\lambda(\vartheta_\lambda^t) \right\|_{\widehat{\mathbf{H}}_\lambda^{-1}(\theta^\star)}$$

$$= \underline{\phi}^{-1} \left( \hat{\mathsf{t}}(\widehat{\theta}_\lambda^t - \vartheta_\lambda^t) \right) e^{\hat{\mathsf{t}}(\vartheta_\lambda^t - \theta^\star)} \left\| \lambda(\widehat{\theta}_\lambda^{t-1} - \vartheta_\lambda^{t-1}) \right\|_{\widehat{\mathbf{H}}_\lambda^{-1}(\theta^\star)}$$

$$= \underline{\phi}^{-1} \left( \hat{\mathsf{t}}(\widehat{\theta}_\lambda^t - \vartheta_\lambda^t) \right) e^{\hat{\mathsf{t}}(\vartheta_\lambda^t - \theta^\star)} \left\| \lambda \widehat{\mathbf{H}}_\lambda^{-1}(\theta^\star)(\widehat{\theta}_\lambda^{t-1} - \vartheta_\lambda^{t-1}) \right\|_{\widehat{\mathbf{H}}_\lambda(\theta^\star)}.$$

Let us detail the recursion. Let $k \leq t$. Then, the following inequality holds, thanks to Lemma 2:

$$\left\|\lambda^k \widehat{\mathbf{H}}_\lambda^{-k}(\theta^\star)(\widehat{\theta}_\lambda^{t-k} - \vartheta_\lambda^{t-k})\right\|_{\widehat{\mathbf{H}}_\lambda(\theta^\star)} \leq \underline{\phi}^{-1}\left(\widehat{\mathsf{t}}(\widehat{\theta}_\lambda^{t-k} - \vartheta_\lambda^{t-k})\right) e^{\widehat{\mathsf{t}}(\vartheta_\lambda^{t-k} - \theta^\star)}$$
$$\left\|\lambda^k \widehat{\mathbf{H}}_\lambda^{-k}(\theta^\star)\left(\nabla\widehat{L}_\lambda(\widehat{\theta}_\lambda^{t-k}) - \nabla\widehat{L}_\lambda(\vartheta_\lambda^{t-k})\right)\right\|_{\widehat{\mathbf{H}}_\lambda^{-1}(\theta^\star)}$$
$$= \underline{\phi}^{-1}\left(\widehat{\mathsf{t}}(\widehat{\theta}_\lambda^{t-k} - \vartheta_\lambda^{t-k})\right) e^{\widehat{\mathsf{t}}(\vartheta_\lambda^{t-k} - \theta^\star)}$$
$$\left\|\lambda^{k+1}\widehat{\mathbf{H}}_\lambda^{-k}(\theta^\star)\left(\widehat{\theta}_\lambda^{t-(k+1)} - \vartheta_\lambda^{t-(k+1)}\right)\right\|_{\widehat{\mathbf{H}}_\lambda^{-1}(\theta^\star)}$$
$$= \underline{\phi}^{-1}\left(\widehat{\mathsf{t}}(\widehat{\theta}_\lambda^{t-k} - \vartheta_\lambda^{t-k})\right) e^{\widehat{\mathsf{t}}(\vartheta_\lambda^{t-k} - \theta^\star)}$$
$$\left\|\lambda^{k+1}\widehat{\mathbf{H}}_\lambda^{-(k+1)}(\theta^\star)\left(\widehat{\theta}_\lambda^{t-(k+1)} - \vartheta_\lambda^{t-(k+1)}\right)\right\|_{\widehat{\mathbf{H}}_\lambda(\theta^\star)}.$$

Thus, unfolding the recursion, we obtain:

$$\left\|\widehat{\theta}_\lambda^t - \vartheta_\lambda^t\right\|_{\widehat{\mathbf{H}}_\lambda(\theta^\star)} \leq \mathsf{P}_\lambda^t \left\|\lambda^t \widehat{\mathbf{H}}_\lambda^{-t}(\theta^\star)\theta^\star\right\|_{\widehat{\mathbf{H}}_\lambda(\theta^\star)},$$
$$\text{with} \quad \mathsf{P}_\lambda^t \stackrel{\text{def.}}{=} \prod_{k=1}^t \underline{\phi}^{-1}\left(\widehat{\mathsf{t}}(\widehat{\theta}_\lambda^k - \vartheta_\lambda^k)\right) e^{\widehat{\mathsf{t}}(\vartheta_\lambda^k - \theta^\star)}. \tag{34}$$

We now use the source condition on $\theta^\star$. Recall that it gives

$$\theta^\star = \mathbf{H}^r(\theta^\star)v,$$

for some $v \in \mathcal{H}$. Thus, we have:

$$\left\|\lambda^t \widehat{\mathbf{H}}_\lambda^{-t}(\theta^\star)\theta^\star\right\|_{\widehat{\mathbf{H}}_\lambda(\theta^\star)} = \left\|\lambda^t \widehat{\mathbf{H}}_\lambda^{-(t-1/2)}(\theta^\star)\mathbf{H}^r(\theta^\star)v\right\| \tag{35}$$
$$\leq \left\|\lambda^t \widehat{\mathbf{H}}_\lambda^{-(t-1/2)}(\theta^\star)\mathbf{H}^r(\theta^\star)\right\| \|v\| \tag{36}$$

We need to distinguish between $r \leq 1$ and $r > 1$ to bound the operator norm

$$\left\|\lambda^t \widehat{\mathbf{H}}_\lambda^{-(t-1/2)}(\theta^\star)\mathbf{H}^r(\theta^\star)\right\|.$$

**Case $r \leq 1$.** We use the following decomposition:

$$\left\|\lambda^t \widehat{\mathbf{H}}_\lambda^{-(t-1/2)}(\theta^\star)\mathbf{H}^r(\theta^\star)\right\| \leq \left\|\lambda^t \widehat{\mathbf{H}}_\lambda^{-(t-1/2)}(\theta^\star)\widehat{\mathbf{H}}_\lambda^r(\theta^\star)\right\| \left\|\widehat{\mathbf{H}}_\lambda^{-r}(\theta^\star)\mathbf{H}^r(\theta^\star)\right\|.$$

The first term is bounded like this:

$$\left\|\lambda^t \widehat{\mathbf{H}}_\lambda^{-(t-1/2)+r}(\theta^\star)\right\| \leq \sup_{\widehat{\sigma}_{\min} < \sigma \leq \mathsf{B}_2^\star} \frac{\lambda^t}{(\sigma + \lambda)^{t-1/2-r}}$$
$$\leq \lambda^s \begin{cases} 1 & \text{if } r + 1/2 < t \\ \mathsf{B}_2^\star + \lambda & \text{if } t = 1 \text{ and } r > 1/2. \end{cases}$$

This illustrates that Tikhonov regularization ($t = 1$) saturates at $r = 1/2$.

For the second term, write

$$\left\|\widehat{\mathbf{H}}_\lambda^{-r}(\theta^\star)\mathbf{H}^r(\theta^\star)\right\| \leq \left\|\widehat{\mathbf{H}}_\lambda^{-r}(\theta^\star)\mathbf{H}_\lambda^r(\theta^\star)\right\|$$

Then, use the Hermitian inequalities of Eq. (67) in Lemma 4, then use the concentration inequalities of Proposition 9. Both can be found in Appendix D. In details:

- If $r \leq 1/2$, use then the concentration inequality of Eq. (64):

$$\left\|\widehat{\mathbf{H}}_\lambda^{-r}(\theta^\star)\mathbf{H}_\lambda^r(\theta^\star)\right\| \leq \left\|\widehat{\mathbf{H}}_\lambda^{-1/2}(\theta^\star)\mathbf{H}_\lambda^{1/2}(\theta^\star)\right\|^{2r}$$
$$\leq 2^{r/2} \text{ if } \mathsf{H}_1.$$

with confidence $1 - \delta$.

- If $r > 1/2$, use the concentration inequality of Eq. (65):

$$\left\|\widehat{\mathbf{H}}_\lambda^{-r}(\theta^\star)\mathbf{H}_\lambda^r(\theta^\star)\right\| \leq \left\|\widehat{\mathbf{H}}_\lambda^{-1}(\theta^\star)\mathbf{H}_\lambda(\theta^\star)\right\|^r$$

$$\leq 2^r \text{ if } \mathsf{H}_{1b} : n \geq 8\frac{\mathsf{B}_2^{\star 2}}{\lambda^2}\log^2\frac{2}{\delta}.$$

All in all, after simplification, the bound on the operator norm when $r \leq 1$ reads

$$\left\|\lambda^t\widehat{\mathbf{H}}_\lambda^{-(t-1/2)}(\theta^\star)\mathbf{H}^r(\theta^\star)\right\| \leq \lambda^s(1 \vee (\mathsf{B}_2^\star + \lambda))2^r \quad \text{if } \begin{cases} \mathsf{H}_1 & \text{when } r \leq 1/2 \\ \mathsf{H}_{1b} & \text{when } r > 1/2, \end{cases} \tag{37}$$

with confidence $1 - \delta$. We now turn to the case $r > 1$.

**Case $r > 1$.** We tackle this case with a different decomposition:

$$\left\|\lambda^t\widehat{\mathbf{H}}_\lambda^{-(t-1/2)}(\theta^\star)\mathbf{H}^r(\theta^\star)\right\| \leq \left\|\lambda^t\widehat{\mathbf{H}}_\lambda^{-(t-1/2)}(\theta^\star)\widehat{\mathbf{H}}^r(\theta^\star)\right\| + \left\|\lambda^t\widehat{\mathbf{H}}_\lambda^{-(t-1/2)}(\theta^\star)(\mathbf{H}^r(\theta^\star) - \widehat{\mathbf{H}}^r(\theta^\star))\right\|$$

Looking at the first term; recalling that $\widehat{\mathbf{H}}(\theta^\star) \leq \mathsf{B}_2^\star$, we have:

$$\left\|\lambda^t\widehat{\mathbf{H}}_\lambda^{-(t-1/2)}(\theta^\star)\widehat{\mathbf{H}}^r(\theta^\star)\right\| \leq \sqrt{\lambda} \sup_{0 < \sigma \leq \mathsf{B}_2^\star}\left(\frac{\lambda}{\lambda+\sigma}\right)^{t-1/2}\sigma^r$$

$$\leq \lambda^s\begin{cases} \frac{r}{(t-1/2)^r} & \text{if } r + 1/2 < t \\ \frac{\mathsf{B}_2^{\star r}}{\left(\mathsf{B}_2^{\star r}+\lambda\right)^{t-1/2}} & \text{otherwise} \end{cases}$$

$$\leq \lambda^s\begin{cases} \frac{r}{(t-1/2)^r} & \text{if } r + 1/2 < t \\ \mathsf{B}_2^{\star r-t+1/2} & \text{otherwise} \end{cases}$$

where we used the computation of Lemma 5. The second term can be upper bounded as follows:

$$\left\|\lambda^t\widehat{\mathbf{H}}_\lambda^{-(t-1/2)}(\theta^\star)(\mathbf{H}^r(\theta^\star) - \widehat{\mathbf{H}}^r(\theta^\star))\right\| \leq \left\|\lambda^t\widehat{\mathbf{H}}_\lambda^{-(t-1/2)}(\theta^\star)\right\|\left\|\mathbf{H}^r(\theta^\star) - \widehat{\mathbf{H}}^r(\theta^\star)\right\|$$

$$\leq w(r)\sqrt{\lambda}\left\|\mathbf{H}(\theta^\star) - \widehat{\mathbf{H}}(\theta^\star)\right\|$$

$$\leq w(r)\frac{\lambda^s}{(t-1/2)^r} \text{ if } \mathsf{H}_2 : n \geq 2\left(1 \vee \left(\frac{2\mathsf{B}_2^\star(t-1/2)^r}{\lambda^{s-1/2}}\right)^2\right)\log\frac{2}{\delta}$$

with confidence $1 - \delta$. We applied Eq. (68) in Lemma 4 on the second inequality, and Eq. (66) in Proposition 9 for the last inequality, both of which can be found in Appendix D. We used:

$$w(r) = r2^{\lfloor r \rfloor + 1}\mathsf{B}_2^{\star r}. \tag{38}$$

Thus, the bound on the operator norm when $r > 1$ reads:

$$\left\|\lambda^t\widehat{\mathbf{H}}_\lambda^{-(t-1/2)}(\theta^\star)\mathbf{H}^r(\theta^\star)\right\| \leq \lambda^s\begin{cases} \frac{w(r)+r}{(t-1/2)^r} & \text{if } r + 1/2 < t \\ \frac{w(r)}{(t-1/2)^r} + \mathsf{B}_2^{\star r-t+1/2} & \text{otherwise} \end{cases} \quad \text{if } \mathsf{H}_2, \tag{39}$$

with confidence $1 - \delta$.

**Gluing things together.** We proceed to the conclusion. Define the following conditions:

$$\mathsf{H}_1 : n \geq 24\frac{\mathsf{B}_2^\star}{\lambda}\log\frac{16\mathsf{B}_2^\star}{\lambda\delta},$$

$$\mathsf{H}_{1b} : n \geq 8\frac{\mathsf{B}_2^{\star 2}}{\lambda^2}\log^2\frac{4}{\delta},$$

$$\mathsf{H}_2 : n \geq 2\left[1 \vee \left(\frac{2\mathsf{B}_2^\star(t-1/2)^r}{\lambda^{s-1/2}}\right)^2\right]\log\frac{4}{\delta},$$

where we replace $\delta$ by $\delta/2$ in order to have bounds with confidence $1 - \delta/2$, so that the overall bound holds with confidence $1 - \delta$ (in fact, $1 - \delta/2$ in the first case). Now assume the following:

$$
\begin{array}{ll}
\mathsf{H}_1 & \text{if } r \leq 1/2, \\
\mathsf{H}_1 + \mathsf{H}_{1b} & \text{if } 1/2 < r \leq 1, \\
\mathsf{H}_1 + \mathsf{H}_2 & \text{if } 1 < r.
\end{array}
$$

Then, we can chain the inequalities of Eqs. (32), (36), (37) and (39). We obtain:

$$
\left\| \widehat{\theta}_\lambda^t - \vartheta_\lambda^t \right\|_{\mathbf{H}(\theta^\star)} \leq \sqrt{2} \, \|v\| \, \mathsf{P}_\lambda^t \lambda^s
\begin{cases}
(1 \vee (\mathsf{B}_2^\star + \lambda)) 2^r & \text{if } r \leq 1, \\
\frac{w(r)+r}{(t-1/2)^r} & \text{if } r > 1 \text{ and } r + 1/2 < t, \\
\frac{w(r)}{(t-1/2)^r} + \mathsf{B}_2^{\star r - t + 1/2} & \text{if } r > 1 \text{ and } r + 1/2 \geq t,
\end{cases}
\tag{40}
$$

with confidence $1 - \delta$. $\qquad\square$

## B.3  Bounding the variance

After bounding the bias, we study the variance term: $\|\vartheta_\lambda^t - \theta^\star\|_{\mathbf{H}(\theta^\star)}$.

**Theorem 3 (Optimal variance of Iterated Tikhonov estimator).** *Let $\delta \in (0, 1]$. Recall the definition of the degrees of freedom $\mathsf{df}_\lambda$. Define the following conditions on the number of samples:*

$$
\mathsf{H}_1 : n \geq 24 \frac{\mathsf{B}_2^\star}{\lambda} \log \frac{16 \mathsf{B}_2^\star}{\lambda \delta},
$$

$$
\mathsf{H}_3 : n \geq 2 \frac{\mathsf{B}_1^{\star 2}}{\lambda \, \mathsf{df}_\lambda} \log \frac{4}{\delta}.
$$

*Then, with probability greater than $1 - \delta$:*

$$
\left\| \vartheta_\lambda^t - \theta^\star \right\|_{\mathbf{H}(\theta^\star)} \leq 4\sqrt{2} t \mathsf{R}_\lambda^t \sqrt{\frac{\mathsf{df}_\lambda}{n}} \cdot \sqrt{\log 2/\delta},
$$

*where we introduced:*

$$
\mathsf{R}_\lambda^t \stackrel{\text{def.}}{=} \prod_{k=1}^{t} \underline{\phi}^{-1} \left( \hat{\mathsf{t}}(\vartheta_\lambda^k - \theta^\star) \right).
$$

*Proof.* The proof begins similarly to the study of the bias term (Theorem 2).

**Changing the norm.** We have the following bound (proof of Theorem 2, Eq. (32)):

$$
\mathsf{H}_1 : n \geq 24 \frac{\mathsf{B}_2^\star}{\lambda} \log \frac{8 \mathsf{B}_2^\star}{\lambda \delta} \implies \left\| \vartheta_\lambda^t - \theta^\star \right\|_{\mathbf{H}(\theta^\star)} \leq \sqrt{2} \left\| \vartheta_\lambda^t - \theta^\star \right\|_{\widehat{\mathbf{H}}_\lambda(\theta^\star)}.
\tag{41}
$$

**Upper bounds on gradient.** To ease the notation, we denote by $\mathsf{a}_k = \underline{\phi}^{-1} \left( \hat{\mathsf{t}}(\vartheta_\lambda^k - \theta^\star) \right)$. We have, thanks to the lower bound on gradient of Eq. (25):

$$
\begin{aligned}
\left\| \vartheta_\lambda^t - \theta^\star \right\|_{\widehat{\mathbf{H}}_\lambda(\theta^\star)} &\leq \mathsf{a}_t \left\| \nabla \widehat{L}_\lambda(\vartheta_\lambda^t) - \nabla \widehat{L}_\lambda(\theta^\star) \right\|_{\widehat{\mathbf{H}}_\lambda^{-1}(\theta^\star)} \\
&= \mathsf{a}_t \left\| \lambda(\vartheta_\lambda^{t-1} - \theta^\star) - \nabla \widehat{L}(\theta^\star) \right\|_{\widehat{\mathbf{H}}_\lambda^{-1}(\theta^\star)} \\
&= \mathsf{a}_t \left\| \lambda \widehat{\mathbf{H}}_\lambda^{-1}(\theta^\star)(\vartheta_\lambda^{t-1} - \theta^\star) \right\|_{\widehat{\mathbf{H}}_\lambda(\theta^\star)} + \mathsf{a}_t \left\| \nabla \widehat{L}(\theta^\star) \right\|_{\widehat{\mathbf{H}}_\lambda^{-1}(\theta^\star)} \\
&\leq \mathsf{a}_t \mathsf{a}_{t-1} \left\| \lambda^2 \widehat{\mathbf{H}}_\lambda^{-2}(\theta^\star)(\vartheta_\lambda^{t-2} - \theta^\star) \right\|_{\widehat{\mathbf{H}}_\lambda(\theta^\star)} + \left[ \mathsf{a}_t + \mathsf{a}_{t-1} \left\| \lambda \widehat{\mathbf{H}}_\lambda^{-1}(\theta^\star) \right\| \right] \left\| \nabla \widehat{L}(\theta^\star) \right\|_{\widehat{\mathbf{H}}_\lambda^{-1}(\theta^\star)}
\end{aligned}
$$

We can unfold the recursion. The first term will disappears thanks to $\vartheta_\lambda^0 = \theta^\star$, and we are left with:

$$\left\|\vartheta_\lambda^t - \theta^\star\right\|_{\widehat{\mathbf{H}}_\lambda(\theta^\star)} \leq \sum_{k=0}^{t-1}\left(\prod_{i=t-k}^{t}\mathsf{a}_i\right)\left\|\lambda^k\widehat{\mathbf{H}}_\lambda^{-k}(\theta^\star)\right\|\left\|\nabla\widehat{L}(\theta^\star)\right\|_{\widehat{\mathbf{H}}_\lambda^{-1}(\theta^\star)} \tag{42}$$

$$\leq \mathsf{R}_\lambda^t\left\|\widehat{\mathbf{H}}_\lambda^{-1/2}(\theta^\star)\mathbf{H}_\lambda^{1/2}(\theta^\star)\right\|\left(\sum_{k=0}^{t-1}\left\|\lambda^k\widehat{\mathbf{H}}_\lambda^{-k}(\theta^\star)\right\|\right)\left\|\nabla\widehat{L}(\theta^\star)\right\|_{\mathbf{H}_\lambda^{-1}(\theta^\star)}, \tag{43}$$

$$\text{where}\quad \mathsf{R}_\lambda^t \overset{\text{def.}}{=} \prod_{k=1}^{t}\underline{\phi}^{-1}\left(\hat{\mathsf{t}}(\vartheta_\lambda^k - \theta^\star)\right). \tag{44}$$

Consider the prefactor of $\|\nabla\widehat{L}(\theta^\star)\|_{\mathbf{H}_\lambda^{-1}(\theta^\star)}$. We will bound $\|\widehat{\mathbf{H}}_\lambda^{-1/2}(\theta^\star)\mathbf{H}_\lambda^{1/2}(\theta^\star)\|$ by $\sqrt{2}$ with the same concentration argument as for the bias. The sum is more difficult to deal with. By computing the supremum of $\sigma \mapsto \lambda^k/(\sigma+\lambda)^k$ we would find that the first $\lfloor t/2\rfloor$ terms have their maximum in $0$. We would end up with a bound for the sum of the order of $t/2$. We rather use the simpler, if not optimal, following bound:

$$\sum_{k=0}^{t-1}\left\|\lambda^k\widehat{\mathbf{H}}_\lambda^{-k}(\theta^\star)\right\| \leq t.$$

It is suboptimal, but of the same order of an exact computation of the operator norm. Thus, we now have:

$$\left\|\vartheta_\lambda^t - \theta^\star\right\|_{\widehat{\mathbf{H}}_\lambda(\theta^\star)} \leq \sqrt{2}t\mathsf{R}_\lambda^t\left\|\nabla\widehat{L}(\theta^\star)\right\|_{\mathbf{H}_\lambda^{-1}(\theta^\star)}\quad\text{when }\mathsf{H}_1 \tag{45}$$

**Bounding the gradient** $\|\nabla\widehat{L}(\theta^\star)\|_{\mathbf{H}_\lambda^{-1}(\theta^\star)}$**.**  We use a plain Bernstein inequality to bound the gradient, as in Proposition 9:

$$\mathbf{H}_\lambda^{-1/2}(\theta^\star)\nabla\widehat{L}(\theta^\star) = \frac{1}{n}\sum_{k=1}^{n}\mathbf{H}_\lambda^{-1/2}(\theta^\star)\nabla\ell_{z_i}(\theta^\star).$$

We have

$$\sup_{z\in\mathsf{Supp}\,\rho}\|\nabla\ell_z(\theta^\star)\|_{\mathbf{H}_\lambda^{-1}(\theta^\star)} \leq \frac{\mathsf{B}_1^\star}{\sqrt{\lambda}},$$

$$\text{and}\quad \mathbb{E}_{z\sim\rho}\left[\|\nabla\ell_z(\theta^\star)\|_{\mathbf{H}_\lambda^{-1}(\theta^\star)}\right]^2 \overset{\text{def.}}{=} \mathsf{df}_\lambda.$$

With confidence $1-\delta$, we now have

$$\left\|\nabla\widehat{L}(\theta^\star)\right\|_{\mathbf{H}_\lambda^{-1}(\theta^\star)} \leq \frac{\mathsf{B}_1^\star}{\sqrt{\lambda}}\frac{2\log 2/\delta}{n} + \sqrt{\mathsf{df}_\lambda\frac{2\log 2/\delta}{n}}.$$

We simplify this equation. Assuming

$$\mathsf{H}_3 : n \geq 2\frac{\mathsf{B}_1^{\star 2}}{\lambda\,\mathsf{df}_\lambda}\log\frac{2}{\delta}$$

we get the bound:

$$\left\|\nabla\widehat{L}(\theta^\star)\right\|_{\mathbf{H}_\lambda^{-1}(\theta^\star)} \leq 2\sqrt{2}\sqrt{\mathsf{df}_\lambda\frac{2\log 2/\delta}{n}}. \tag{46}$$

**Gluing things together.**  All in all, we can glue together the inequalities in Eqs. (41), (45) and (46). We obtain:

$$\left\|\vartheta_\lambda^t - \theta^\star\right\|_{\mathbf{H}(\theta^\star)} \leq 4\sqrt{2}t\mathsf{R}_\lambda^t\sqrt{\frac{\mathsf{df}_\lambda}{n}}\cdot\sqrt{\log 2/\delta}\quad\text{when }\mathsf{H}_1 + \mathsf{H}_3$$

with confidence $1-2\delta$. We obtain the statement of the theorem by replacing $\delta$ with $\delta/2$, so that the result holds with confidence $1-\delta$. $\qquad\qquad\square$

### B.4 Conditions for non-exponentials prefactors

The prefactors $\mathsf{P}_\lambda^t$ and $\mathsf{R}_\lambda^t$ are hard to bound; they can depend exponentially on $\|\theta^\star\|$ in the worst case [1]. The purpose of this section is to give sufficient conditions on the number of samples $n$ for those quantities to turn constant. The key quantity to compare to is the *Dikin radius* [1, 14].

**Definition 7 (Dikin radius).** *For $\theta \in \mathcal{H}$ and $\lambda > 0$, define $\mathsf{r}_\lambda(\theta)$ s.t*

$$\frac{1}{\mathsf{r}_\lambda(\theta)} = \sup_{z \in \mathsf{Supp}\,\rho} \sup_{g \in \phi(z)} \|g\|_{\mathbf{H}_\lambda^{-1}(\theta)}. \tag{47}$$

The inverse of the Dikin radius can be upper bounded by $R/\sqrt{\lambda}$. However, we prefer keeping bounds in $\mathsf{r}_\lambda^\star$. Indeed, they take into account the geometry of the loss function around the optimum, and are thus much more precise.

Note that in the following, we might be content with the *empirical* Dikin radius $\widehat{\mathsf{r}_\lambda(\theta)}$, ie. replacing $\rho$ by $\hat{\rho}$ in the previous definition. So as not to ladden the notations and have something independant of the sampling, we use the fact that $\mathsf{Supp}\,\hat{\rho} \subset \mathsf{Supp}\,\rho$ to ensure that:

$$\frac{1}{\widehat{\mathsf{r}_\lambda(\theta)}} \leq \frac{1}{\mathsf{r}_\lambda(\theta)} \quad \text{and} \quad \hat{\mathsf{t}}(\cdot) \leq \mathsf{t}(\cdot).$$

Finally, we will use the following notation:

$$\mathsf{r}_\lambda^\star \overset{\text{def.}}{=} \mathsf{r}_\lambda(\theta^\star). \tag{48}$$

#### B.4.1 Pefactor of the variance

We first proceed with the prefactor of the variance $\mathsf{R}_\lambda^t$.

**Proposition 4 (Constant prefactor for the variance).** *The following condition:*

$$\mathsf{H}_4 : n \geq 8(et)^2 \left(4 \vee C^2 t^2\right) \frac{\mathsf{df}_\lambda}{\mathsf{r}_\lambda^\star} \log 2/\delta, \tag{49}$$

*where $C \leq 0.8$ is a constant, is sufficient to guarantee that*

$$\mathsf{R}_\lambda^t \overset{\text{def.}}{=} \prod_{k=1}^t \underline{\phi}^{-1}\left(\hat{\mathsf{t}}(\vartheta_\lambda^k - \theta^\star)\right) \leq e. \tag{50}$$

*Proof.*

**A first bound.** Note that:

$$\mathsf{t}(\vartheta_\lambda^t - \theta^\star) = \sup_{z \in \mathsf{Supp}\,\rho} \sup_{g \in \phi(z)} \left|g \cdot (\vartheta_\lambda^t - \theta^\star)\right|$$

$$\leq \sup_{z \in \mathsf{Supp}\,\rho} \sup_{g \in \phi(z)} \|g\|_{\widehat{\mathbf{H}}_\lambda^{-1}(\theta^\star)} \left\|\vartheta_\lambda^t - \theta^\star\right\|_{\widehat{\mathbf{H}}_\lambda(\theta^\star)},$$

which gives us a bound we will use multiple times:

$$\mathsf{t}(\vartheta_\lambda^t - \theta^\star) \leq \frac{\|\vartheta_\lambda^t - \theta^\star\|_{\widehat{\mathbf{H}}_\lambda(\theta^\star)}}{\mathsf{r}_\lambda^\star}. \tag{51}$$

We simply used the definition of the Dikin radius in Eq. (47).

We now use an upper bound of the numerator, available in the proof of Theorem 3:

$$\mathsf{t}(\vartheta_\lambda^t - \theta^\star) \leq \mathsf{R}_\lambda^t \left[2\sqrt{2}t\sqrt{\log 2/\delta}\right] \sqrt{\frac{\mathsf{df}_\lambda}{n\mathsf{r}_\lambda^\star}}$$

$$\iff \mathsf{t}(\vartheta_\lambda^t - \theta^\star)\underline{\phi}\left(\mathsf{t}(\vartheta_\lambda^t - \theta^\star)\right) \leq \mathsf{R}_\lambda^{t-1}\left[2\sqrt{2}t\sqrt{\log 2/\delta}\right] \sqrt{\frac{\mathsf{df}_\lambda}{n\mathsf{r}_\lambda^\star}} \overset{\text{def.}}{=} X_{t-1}.$$

Now, using the fact that $x\underline{\phi}(x) = 1 - e^{-x}$, we get that

$$\mathsf{t}(\vartheta_\lambda^t - \theta^\star) \leq -\log(1 - X_{t-1})$$

$$\mathsf{a}_t \overset{\text{def.}}{=} \underline{\phi}^{-1}\left(\mathsf{t}(\vartheta_\lambda^t - \theta^\star)\right) \leq -X_{t-1}^{-1}\log(1 - X_{t-1}) \overset{\text{def.}}{=} h(X_{t-1}). \tag{52}$$

**Recursion hypotheses.** The idea is to ensure:

1. $X_{k-1} \leq 1/2$ so that
$$h(X_{k-1}) \leq 1 + CX_{k-1}$$
   with $C$ a numeric constant s.t $h(1/2) = 1 + C/2$, which implies that $C \leq 0.8$. We are simply upper bounding $h$ which is convex on $[0, 1/2]$.

2. $\mathsf{a}_k \leq 1 + 1/t$ for all $k \leq t$, so that we can have:
$$\mathsf{R}_\lambda^t = \prod_{k=1}^t \mathsf{a}_k = \exp \sum_{k=1}^t \log(\mathsf{a}_k)$$
$$\leq \exp \sum_{k=1}^t \log(1 + 1/t) \leq e.$$

**Recursion.** Set $k = 1$. Then $\mathsf{R}_\lambda^0 = 1$ and to have
$$X_0 \leq 1/2 \quad \text{that is} \quad \left[ 2\sqrt{2}t\sqrt{\log 2/\delta} \right] \sqrt{\frac{\mathsf{df}_\lambda}{n\mathsf{r}_\lambda^\star}} \leq \frac{1}{2},$$

it is sufficient to have
$$n \geq N_0 \overset{\text{def.}}{=} 32t^2 \frac{\mathsf{df}_\lambda}{\mathsf{r}_\lambda^\star} \log 2/\delta.$$

We want to enforce
$$\mathsf{a}_1 \leq 1 + 1/t.$$

A sufficient condition is
$$h(X_0) \leq 1 + CX_0 \leq 1 + 1/t \impliedby X_0 \leq 1/tC$$
$$\impliedby n \geq N_0' \overset{\text{def.}}{=} 8t^4 C^2 \frac{\mathsf{df}_\lambda}{\mathsf{r}_\lambda^\star} \log 2/\delta.$$

Now, let $k < n$. Assume the two conditions hold at step $k - 1$. Then, $\mathsf{R}_\lambda^{k-1} \leq e$ and
$$n \geq N_{k-1} \overset{\text{def.}}{=} 32(et)^2 \frac{\mathsf{df}_\lambda}{\mathsf{r}_\lambda^\star} \log 2/\delta \quad \text{implies} \quad X_{k-1} \leq \frac{1}{2}.$$

Likewise,
$$n \geq N_{k-1}' \overset{\text{def.}}{=} 8t^4 (Ce)^2 \frac{\mathsf{df}_\lambda}{\mathsf{r}_\lambda^\star} \log 2/\delta$$

gives
$$X_{k-1} \leq 1/tC, \quad \text{so that} \quad \mathsf{a}_k \leq 1 + 1/k.$$

**Conclusion.** All in all, requiring
$$\mathsf{H}_4 : n \geq 8(et)^2 \left( 4 \vee C^2 t^2 \right) \frac{\mathsf{df}_\lambda}{\mathsf{r}_\lambda^\star} \log 2/\delta$$

is sufficient to have $\mathsf{R}_\lambda^k \leq e$, for any $k \leq t$. $\qquad\qquad\square$

### B.4.2 Pefactor of the bias

The prefactor of the bias can be treated similarly. The only difficulty comes from the large number of subcases. Remember from Theorem 2 that we have, with appropriate hypotheses,

$$\left\|\widehat{\theta}_\lambda^t - \vartheta_\lambda^t\right\|_{\widehat{\mathbf{H}}_\lambda(\theta^\star)} \leq \mathsf{P}_\lambda^t \mathsf{T}(r, t)\lambda^s, \quad \text{with } s = (r + 1/2) \wedge t. \tag{53}$$

**Proposition 5 (Constant prefactor for the bias).** *Assume* $\mathsf{H}_4$ *and*

$$\mathsf{H}_5 : \lambda \leq \mathsf{L} \stackrel{\text{def.}}{=} \left[e^{t+2}\mathsf{T}(r, t)(2 \wedge Ct)\right]^{-1/(r+1/2\wedge 1)}.$$

*Then*

$$\mathsf{P}_\lambda^t \leq e^{t+2}.$$

*Proof.* The proof is almost identical to the proof of Proposition 4. Let us simply point out the differences. We will drop the dependance of $\mathsf{T}$ on $r, t$ in the notation for simplicity.

**A first bound.** Here, we have that:

$$\mathsf{t}\left(\widehat{\theta}_\lambda^k - \vartheta_\lambda^k\right) \leq \frac{\left\|\widehat{\theta}_\lambda^k - \vartheta_\lambda^k\right\|_{\widehat{\mathbf{H}}_\lambda(\theta^\star)}}{\mathsf{r}_\lambda^\star} \leq \frac{\mathsf{P}_\lambda^t \mathsf{T}\lambda^s}{\mathsf{r}_\lambda^\star}, \quad \text{with } s = r + 1/2 \wedge k. \tag{54}$$

We used Eq. (53) in the second inequality. Recall the definition

$$\mathsf{P}_\lambda^t \stackrel{\text{def.}}{=} \prod_{k=1}^{t} \underline{\phi}^{-1}\left(\widehat{\mathsf{t}}(\widehat{\theta}_\lambda^k - \vartheta_\lambda^k)\right) e^{\widehat{\mathsf{t}}(\vartheta_\lambda^k - \theta^\star)}.$$

Thanks to $\mathsf{H}_4$, we have $\mathsf{R}_\lambda^t \leq e$. Specifically, noting that $\underline{\phi}^{-1}(x) \geq x$, we have from the proof of Proposition 4:

$$1 + 1/t \geq \underline{\phi}^{-1}\left(\mathsf{t}(\vartheta_\lambda^k - \theta^\star)\right) \geq \mathsf{t}(\vartheta_\lambda^k - \theta^\star) \implies \prod_{k=1}^{t} e^{\mathsf{t}(\vartheta_\lambda^k - \theta^\star)} \leq e^{t+1}.$$

Thus, we have that

$$\mathsf{P}_\lambda^k \leq \underbrace{\prod_{i=1}^{k} \underline{\phi}^{-1}\left(\mathsf{t}(\widehat{\theta}_\lambda^i - \vartheta_\lambda^i)\right)}_{\mathsf{Q}_\lambda^k} e^{t+1}.$$

Dividing both sides in the Eq. (54) with $\underline{\phi}^{-1}\left(\mathsf{t}(\widehat{\theta}_\lambda^k - \vartheta_\lambda^k)\right)$, we obtain that

$$\mathsf{t}\left(\widehat{\theta}_\lambda^k - \vartheta_\lambda^k\right) \underline{\phi}\left(\mathsf{t}(\widehat{\theta}_\lambda^k - \vartheta_\lambda^k)\right) \leq \frac{\mathsf{Q}_\lambda^{k-1}\mathsf{T}e^{t+1}\lambda^s}{\mathsf{r}_\lambda^\star},$$

and we can apply the same reasoning as for the variance. Using $t\underline{\phi}(t) = 1 - e^{-t}$, we have

$$\mathsf{a}_k \stackrel{\text{def.}}{=} \underline{\phi}^{-1}\left(\mathsf{t}(\widehat{\theta}_\lambda^k - \vartheta_\lambda^k)\right) \leq -X_{k-1}^{-1}\log(1 - X_{k-1})$$

$$X_{k-1} \stackrel{\text{def.}}{=} \mathsf{Q}_\lambda^{k-1}\mathsf{T}e^{t+1}\lambda^s/\mathsf{r}_\lambda^\star.$$

**Recursion.** We then do the exact same reasoning to the variance, that is require at each step $X_{k-1} \leq 1/2$ and $\mathsf{a}_k \leq 1 + 1/t$. Here, this amounts to require

$$\lambda \leq \mathsf{L}_s \stackrel{\text{def.}}{=} \left[e^{t+2}\mathsf{T}(2 \wedge Ct)\right]^{-1/s}.$$

The $\mathsf{L}_s$ is increasing with $s$. So

$$\forall k \leq t, \quad \mathsf{L}_s \leq \mathsf{L}_{r+1/2\wedge 1} \stackrel{\text{def.}}{=} \mathsf{L}, \quad \text{with } s = r + 1/2 \wedge k.$$

Table 2: Hypotheses needed to bound the bias and the variance, depending on the source condition parameter $r$.

| Source condition | Bias | Variance | Numerical prefactors |
|---|---|---|---|
| $0 < r \leq 1/2$ | $\mathsf{H}_1$ | | |
| $1/2 < r < 1$ | $\mathsf{H}_1 + \mathsf{H}_{1b}$ | $\mathsf{H}_1 + \mathsf{H}_3$ | $\mathsf{H}_4 + \mathsf{H}_5$ |
| $r \geq 1$ | $\mathsf{H}_1 + \mathsf{H}_2$ | | |

**Conclusion.** Requiring

$$\mathsf{H}_5 : \lambda \leq \mathsf{L} \stackrel{\text{def.}}{=} \left[ e^{t+2} \mathsf{T}(2 \wedge Ct) \right]^{-1/(r+1/2 \wedge 1)}$$

is sufficient to ensure $\mathsf{Q}_\lambda^t \leq e$, so that $\mathsf{P}_\lambda^t \leq e^{t+2}$. $\qquad\square$

### B.5 Optimal rates for IT estimator

The bound on the bias and the variance holds if the number of samples is "high enough". The purpose of next proposition is to merge all these hypotheses together. Precisely, the hypotheses requires in each regime are summed up in Table 2.

**Proposition 6 (Satisfying the hypotheses $\mathsf{H}_{1-5}$ with bounds on $n$ and $\lambda$).** *The following relations hold:*

$$n \geq \mathsf{N}_0 \stackrel{\text{def.}}{=} \frac{2}{\lambda} \left[ 12\mathsf{B}_2^\star \vee \frac{\mathsf{B}_1^{\star 2}}{\mathsf{df}_\lambda} \right] \log \frac{4}{\delta} \left[ 1 \vee \frac{4\mathsf{B}_2^\star}{\lambda} \right] \implies \mathsf{H}_1 + \mathsf{H}_3,$$

$$n \geq \mathsf{N}_{1/2} \stackrel{\text{def.}}{=} \frac{2}{\lambda} \left[ 12\mathsf{B}_2^\star \vee \frac{\mathsf{B}_1^{\star 2}}{\mathsf{df}_\lambda} \vee \frac{4\mathsf{B}_2^\star}{\lambda} \right] \log^2 \frac{4}{\delta} \left[ 1 \vee \frac{4\mathsf{B}_2^\star}{\lambda} \right] \implies \mathsf{H}_1 + \mathsf{H}_{1b} + \mathsf{H}_3,$$

$$n \geq \mathsf{N}_1 \stackrel{\text{def.}}{=} \frac{2}{\lambda} \left[ 12\mathsf{B}_2^\star \vee \frac{\mathsf{B}_1^{\star 2}}{\mathsf{df}_\lambda} \vee \lambda \vee \left( \frac{2\mathsf{B}_2^\star (t-1/2)^r}{\lambda^{r-1/2}} \right)^2 \right] \log \frac{4}{\delta} \left[ 1 \vee \frac{4\mathsf{B}_2^\star}{\lambda} \right] \implies \mathsf{H}_1 + \mathsf{H}_2 + \mathsf{H}_3,$$

$$n \geq \overline{\mathsf{N}} \stackrel{\text{def.}}{=} 8(et)^2 \left( 4 \vee C^2 t^2 \right) \frac{\mathsf{df}_\lambda}{\mathsf{r}_\lambda^\star} \log 2/\delta \implies \mathsf{H}_4,$$

$$\lambda \leq \mathsf{L} \stackrel{\text{def.}}{=} \left[ e^{t+2} \mathsf{T}(r,t)(2 \wedge Ct) \right]^{-1/(r+1/2 \wedge 1)} \implies \mathsf{H}_5.$$

*Recall that $\mathsf{T}$ is defined in Theorem 2. Moreover, having*

$$\lambda = Kn^{-\frac{\alpha}{1+\alpha(2r+1)}}$$

*with $K$ a constant not depending on $n$ make all these conditions possible.*

*Proof.* The expression of the constant boils down to taking the maximum of each expression. Recall that:

- $\mathsf{H}_1, \mathsf{H}_{1b}, \mathsf{H}_2$ are defined in Theorem 2;

- $\mathsf{H}_3$ is defined in Theorem 3;

- $\mathsf{H}_4$ is defined in Proposition 4;

- $\mathsf{H}_5$ is defined in Proposition 5.

About the fact they are attainable, we need to check that the power of $n$ is smaller than 1, in order that

$$\exists n, \quad n \geq \mathsf{N}(\lambda) \quad \text{and} \quad \lambda = Kn^{-\frac{\alpha}{1+\alpha(2r+1)}},$$

with $\mathsf{N}(\lambda)$ chosen among $\{\mathsf{N}_0, \mathsf{N}_{1/2}, \mathsf{N}_1, \overline{\mathsf{N}}\}$. In the following, $\sim$ denotes equality up to log factors between two quantities. Recall that $\mathsf{s}\lambda^{-1/\alpha} \leq \mathsf{df}_\lambda \leq \mathsf{S}\lambda^{-1/\alpha}$, and assume

$$\lambda = Kn^{-\frac{\alpha}{1+\alpha(2r+1)}}$$

for some $K$ a positive constant.

- When $r \leq 1/2$, $\mathsf{N}_0 \sim \lambda^{-(1+1/\alpha)} \sim n^{\frac{1+\alpha}{1+\alpha(2r+1)}}$ and $\frac{1+\alpha}{1+\alpha(2r+1)} < 1$.

- When $1/2 < r \leq 1$, $\mathsf{N}_{1/2} \sim \lambda^{-2} \sim n^{\frac{2\alpha}{1+\alpha(2r+1)}}$ and $\frac{2\alpha}{1+\alpha(2r+1)} < 1$ as $\alpha(2r+1) > 2\alpha$.

- Finally, when $r > 1$, $\mathsf{N}_1 \sim \lambda^{-2r} \sim n^{\frac{\alpha(2r)}{1+\alpha(2r+1)}}$ and $\alpha(2r) < 1 + \alpha(2r+1)$.

- For $\overline{\mathsf{N}}$, use the upper bound $\frac{\mathsf{df}_\lambda}{\mathsf{r}_\lambda^\star} \leq \mathsf{S}R\lambda^{-(1/\alpha+1/2)}$. Then, $\overline{\mathsf{N}} \sim n^{\frac{\alpha+2}{2(1+\alpha(2r+1))}}$ and $\frac{\alpha+2}{2(1+\alpha(2r+1))} = 1 - \frac{\alpha(4r+1)}{\alpha(4r+2)+2} \leq 1$.

$\square$

Having bounded the bias and the variance of the estimator, we are now in shape to state our main result.

**Theorem 4 (Optimal rates of IT estimator).** *Let $\delta \in (0,1]$, $\lambda > 0$ and choose $n$ so that $\mathsf{H}_3$ and the following holds:*

$$
\begin{array}{ll}
\mathsf{H}_1 & \text{if } r \leq 1/2, \\
\mathsf{H}_1 + \mathsf{H}_{1b} & \text{if } 1/2 < r \leq 1, \\
\mathsf{H}_1 + \mathsf{H}_2 & \text{if } r > 1.
\end{array}
$$

*Then we can bound the excess risk with probability greater than $1 - \delta$ as*

$$
L(\widehat{\theta}_\lambda^t) - L(\theta^\star) \leq \mathsf{C}_{\mathrm{bias}}\lambda^{2s} + \mathsf{C}_{\mathrm{var}}\frac{\mathsf{df}_\lambda}{n}, \quad \text{with } s = (r+1/2) \wedge t.
$$

*If we further assume that the capacity condition holds and that the estimator does not saturate, that is $t \geq r + 1/2$, then setting*

$$
\lambda = \left[ \left( \frac{\mathsf{C}_{\mathrm{var}}}{\mathsf{C}_{\mathrm{bias}}} \right)^2 \mathsf{S} \right]^{\frac{\alpha}{1+\alpha(2r+1)}} n^{-\frac{\alpha}{1+\alpha(2r+1)}}
$$

*makes the following holds with confidence $2\delta$:*

$$
L(\widehat{\theta}_\lambda^t) - L(\theta^\star) \leq 2 \left[ \left( \frac{\mathsf{C}_{\mathrm{var}}}{\mathsf{C}_{\mathrm{bias}}} \right)^2 \mathsf{S} \right]^{\frac{\alpha(2r+1)}{1+\alpha(2r+1)}} n^{-\frac{\alpha(2r+1)}{1+\alpha(2r+1)}},
$$

*where the constants $\mathsf{C}_{\mathrm{bias}}, \mathsf{C}_{\mathrm{var}}$ are bounded by quantities only depending on $r, t, \mathsf{B}_2^\star, \delta$ as soon as hypotheses $\mathsf{H}_4$ and $\mathsf{H}_5$ are satisfied.*

*Proof.*

**Decomposition of the risk.** We use the decomposition of the risk:

$$
L(\widehat{\theta}_\lambda^t) - L(\theta^\star) \leq \Psi\left( \mathsf{t}(\widehat{\theta}_\lambda^t - \theta^\star) \right) \left\| \widehat{\theta}_\lambda^t - \theta^\star \right\|_{\mathbf{H}(\theta^\star)}^2
$$

$$
\leq 2\Psi\left( \mathsf{t}(\widehat{\theta}_\lambda^t - \vartheta_\lambda^t) + \mathsf{t}(\vartheta_\lambda^t - \theta^\star) \right) \left[ \left\| \widehat{\theta}_\lambda^t - \vartheta_\lambda^t \right\|_{\mathbf{H}(\theta^\star)}^2 + \left\| \vartheta_\lambda^t - \theta^\star \right\|_{\mathbf{H}(\theta^\star)}^2 \right],
$$

where we applied Proposition 3, and used that $(a+b)^2 \leq 2(a^2+b^2)$.

**Bias and variance prefactors.** We introduce the following quantities:

$$
\mathsf{C}_{\mathrm{bias}}^2 = 2\Psi\left( \mathsf{t}(\widehat{\theta}_\lambda^t - \vartheta_\lambda^t) + \mathsf{t}(\vartheta_\lambda^t - \theta^\star) \right) \mathsf{T}(r,t)\mathsf{P}_\lambda^t,
$$

$$
\mathsf{C}_{\mathrm{var}}^2 = 2\Psi\left( \mathsf{t}(\widehat{\theta}_\lambda^t - \vartheta_\lambda^t) + \mathsf{t}(\vartheta_\lambda^t - \theta^\star) \right) \left[ 4\sqrt{2}t\mathsf{R}_\lambda^t \sqrt{\log 2/\delta} \right],
$$

where $\mathsf{P}_\lambda^t, \mathsf{R}_\lambda^t$ are defined in Theorems 2 and 3 respectively. Then the bound on the excess risk reads:

$$L(\widehat{\theta}_\lambda^t) - L(\theta^\star) \leq \mathsf{C}_{\mathrm{bias}} \begin{cases} \lambda^{2r+1} & \text{if } r + 1/2 \leq t \\ \lambda^{2t} & \text{otherwise} \end{cases} + \mathsf{C}_{\mathrm{var}} \frac{\mathsf{df}_\lambda}{n}$$

with confidence $2\delta$ with the appropriate hypothesis $\mathsf{H}_1, \mathsf{H}_2$ or $\mathsf{H}_3$, depending on $r$, see Table 2.

**Optimal $\lambda$.** Further assume $t \geq r + 1/2$ and the capacity condition holds with parameters $\mathsf{S}, \alpha$. Then, setting:

$$\lambda^{\frac{1+\alpha(2r+1)}{\alpha}} = \left( \frac{\mathsf{C}_{\mathrm{var}}}{\mathsf{C}_{\mathrm{bias}}} \right)^2 \frac{\mathsf{S}}{n} \iff \lambda = \left[ \left( \frac{\mathsf{C}_{\mathrm{var}}}{\mathsf{C}_{\mathrm{bias}}} \right)^2 \mathsf{S} \right]^{\frac{\alpha}{1+\alpha(2r+1)}} n^{-\frac{\alpha}{1+\alpha(2r+1)}},$$

makes the following bound holds with probability $1 - 2\delta$:

$$L(\widehat{\theta}_\lambda^t) - L(\theta^\star) \leq 2 \left[ \left( \frac{\mathsf{C}_{\mathrm{var}}}{\mathsf{C}_{\mathrm{bias}}} \right)^2 \mathsf{S} \right]^{\frac{\alpha(2r+1)}{1+\alpha(2r+1)}} n^{-\frac{\alpha(2r+1)}{1+\alpha(2r+1)}}.$$

**Explicit prefactors.** Assume Hyp. $\mathsf{H}_4$ and $\mathsf{H}_5$ hold, and $\lambda \leq \mathsf{B}_2^\star$. Then the quantities $\mathsf{C}_{\mathrm{bias}}, \mathsf{C}_{\mathrm{var}}$ only depend on $r, t$ up to the term $\Psi\left( \mathsf{t}(\widehat{\theta}_\lambda^t - \vartheta_\lambda^t) + \mathsf{t}(\vartheta_\lambda^t - \theta^\star) \right)$. Noting that:

$$1 + 1/t \geq \underline{\phi}^{-1}(x) \geq x \quad \text{implies} \quad 1 + 1/t \geq x,$$

and $\Psi$ is increasing we have

$$\Psi\left( \mathsf{t}(\widehat{\theta}_\lambda^t - \vartheta_\lambda^t) + \mathsf{t}(\vartheta_\lambda^t - \theta^\star) \right) \leq \Psi(4) \leq 4.$$

In the end $\mathsf{C}_{\mathrm{bias}}, \mathsf{C}_{\mathrm{var}}$ only depend on $r, t$ and the parameters of the problem:

$$\begin{aligned} \mathsf{C}_{\mathrm{bias}}^2 &\leq 8\mathsf{T}(r,t)e^{t+2} \\ \mathsf{C}_{\mathrm{var}}^2 &\leq 32te\sqrt{\log 2/\delta}, \end{aligned} \tag{55}$$

where $\mathsf{T}(r,t)$ was introduced previously in Theorem 2:

$$\mathsf{T}(r,t) = \begin{cases} \|v\| \, (1 \vee (\mathsf{B}_2^\star + \lambda))2^r & \text{if } r \leq 1, \\ \|v\| \frac{w(r)+r}{(t-1/2)^r} & \text{if } r > 1 \text{ and } r + 1/2 < t, \\ \|v\| \frac{w(r)}{(t-1/2)^r} + \mathsf{B}_2^{\star r-t+1/2} & \text{if } r > 1 \text{ and } r + 1/2 \geq t. \end{cases}$$

**Proof of Theorem 1 in the paper.** We took the maximum on the lower bounds on the samples to simplify the result in the main body. Simply define:

$$\mathsf{N} = \overline{\mathsf{N}} \vee \begin{cases} \mathsf{N}_0 & \text{if } r \leq 1/2 \\ \mathsf{N}_{1/2} & \text{if } 1/2 < r < 1 \\ \mathsf{N}_1 & \text{otherwise} \end{cases}$$

$$\text{and} \quad \mathsf{C}_{\mathrm{risk}} = \left[ \left( \frac{\mathsf{C}_{\mathrm{var}}}{\mathsf{C}_{\mathrm{bias}}} \right)^2 \mathsf{S} \right]^{\frac{\alpha}{1+\alpha(2r+1)}}.$$

Again, we highlight that the observation made in Proposition 6 is key to ensure that these constants are attainable, in the sense that they are not in contradiction with the optimal rate in $n$. $\qquad \square$

## C  Statistical guarantees with inexact solvers

This section is devoted to finding a rule on the tolerance enforced at each step of the proximal sequence. Given a tolerance $\epsilon$, we look for $\bar{\epsilon}_1, \ldots, \bar{\epsilon}_n$, the tolerance to ensure at each proximal step. It leads to Proposition 1 in the main body of the article.

**Important remark on the notation.**  So as to simplify the notation, we drop the hat ^ on the loss function. That is, we simply take a loss function $L$ assumed to be GSC. In practice, this function is of course the empirical loss $\widehat{L}$. We denote with a bar ⁻ the quantity we compute at each step, and whose aim is to approximate the estimator of $L$.

**Tikhonov regularization.**  For a GSC function $L$, we define:

$$\theta_\mu^1 = \mathsf{prox}_{L/\mu}(0) = \arg\min_\theta L_\mu(\theta), \qquad\qquad L_\mu(\theta) \overset{\text{def.}}{=} L(\theta) + \frac{\mu}{2}\|\theta\|^2$$

$$\theta_\mu^{k+1} = \mathsf{prox}_{L/\mu}(\theta_\lambda^k) = \arg\min_\theta L_\mu^{\lambda,k}(\theta), \qquad L_\mu^{\lambda,k}(\theta) \overset{\text{def.}}{=} L(\theta) + \frac{\mu}{2}\left\|\theta - \theta_\lambda^k\right\|^2$$

$$\overline{\theta}_\mu^{k+1} = \mathsf{prox}_{L/\mu}(\overline{\theta}_\lambda^k) = \arg\min_\theta \overline{L}_\mu^{\lambda,k}(\theta), \qquad \overline{L}_\mu^{\lambda,k}(\theta) \overset{\text{def.}}{=} L(\theta) + \frac{\mu}{2}\left\|\theta - \overline{\theta}_\lambda^k\right\|$$

so that we can refer easily to the function which has to be minimized when evaluating the proximal operator.

## C.1   Definitions

We use the following notations for the *Newton decrement*:

- The theoretical quantity writes:

$$\nu_\mu^{\lambda,k}(\theta) = \left\|\nabla L_\mu^{\lambda,k}(\theta)\right\|_{\mathbf{H}_\lambda^{-1}(\theta)} = \left\|\nabla L(\theta) + \mu(\theta - \theta_\lambda^{\lambda,k})\right\|_{\mathbf{H}_\mu^{-1}(\theta)};$$

- The normalized Newton decrement is defined with:

$$\widetilde{\nu}_\lambda^{k-1}(\theta) = \frac{\nu_\lambda^{k-1}(\theta)}{\mathsf{r}_\lambda(\theta)};$$

- The quantity we compute is:

$$\overline{\nu}_\mu^{\lambda,k}(\theta) = \left\|\nabla L_\mu^{\lambda,k}(x)\right\|_{\mathbf{H}_\lambda^{-1}(\theta)} = \left\|\nabla L(\theta) + \mu(\theta - \overline{\theta}_\lambda^{\lambda,k})\right\|_{\mathbf{H}_\mu^{-1}(\theta)}.$$

We also recall some definition and properties. $R$ is defined with

$$R = \sup_{z\in\mathsf{Supp}\,\rho}\sup_{g\in\phi(z)} \|g\| \quad \text{so that} \quad \mathsf{r}_\lambda(\theta) \geq R/\sqrt{\lambda}$$

and $\mathsf{r}_\lambda(\theta)$ is given in Definition 7. The Dikin ellipsoid, as in [14], reads

$$\forall \mathsf{c} \in \mathbb{R}, \quad \mathsf{D}_\lambda^{k-1}(\mathsf{c}) = \left\{\theta \in \mathcal{H}; \widetilde{\nu}_\lambda^{k-1}(\theta) \leq \mathsf{c}\right\}.$$

We provide a short lemma to show how controlling the *normalized* Newton decrement enables to control quantities depending on $\mathsf{t}$.

**Lemma 3 (Localization properties with the Newton decrement).** *Let $k \leq t$ and $\mathsf{c} > 0$. Assume*

$$\overline{\theta}_\lambda^k \in \mathsf{D}_\lambda^{k-1}(\mathsf{c}), \quad \text{that is} \quad \widetilde{\nu}_\lambda^{k-1}\left(\overline{\theta}_\lambda^k\right) \leq \mathsf{c}.$$

*Then, we have*

$$\underline{\phi}^{-1}\left(\mathsf{t}(\overline{\theta}_\lambda^k - \theta_\lambda^k)\right) \leq -\frac{1}{\mathsf{c}}\log(1-\mathsf{c}) \overset{\text{def.}}{=} \kappa_\mathsf{c}. \tag{56}$$

*Proof.* The proof combines inequalities we already used, replacing the normalized gradient with the Newton decrement. Recall Eq. (51), which states that

$$\mathsf{t}(\overline{\theta}_\lambda^k - \theta_\lambda^k) \leq \frac{\left\|\overline{\theta}_\lambda^k - \theta_\lambda^k\right\|_{\widehat{\mathbf{H}}_\lambda(\overline{\theta}_\lambda^k)}}{\mathsf{r}_\lambda(\overline{\theta}_\lambda^k)}. \tag{57}$$

Using the lower bound on gradient of Lemma 2 gives

$$\left\|\overline{\theta}_\lambda^k - \theta_\lambda^k\right\|_{\widehat{\mathbf{H}}_\lambda(\overline{\theta}_\lambda^k)} \leq \underline{\phi}^{-1}(\mathsf{t}(\overline{\theta}_\lambda^k - \theta_\lambda^k)) \left\|\nabla L_\lambda^{k-1}(\overline{\theta}_\lambda^k)\right\|_{\mathbf{H}_\lambda^{-1}(\overline{\theta}_\lambda^k)},$$

and using the definition of the Newton decrement in the previous equation gives

$$\left\|\overline{\theta}_\lambda^k - \theta_\lambda^k\right\|_{\widehat{\mathbf{H}}_\lambda(\overline{\theta}_\lambda^k)} \leq \underline{\phi}^{-1}(\mathsf{t}(\overline{\theta}_\lambda^k - \theta_\lambda^k))\nu_\lambda^{k-1}\left(\overline{\theta}_\lambda^k\right). \tag{58}$$

Plugging Eq. (58) in Eq. (57) implies

$$\underline{\phi}(\mathsf{t}(\overline{\theta}_\lambda^k - \theta_\lambda^k))\,\mathsf{t}(\overline{\theta}_\lambda^k - \theta_\lambda^k) \leq \frac{\nu_\lambda^{k-1}\left(\overline{\theta}_\lambda^k\right)}{\mathsf{r}_\lambda(\overline{\theta}_\lambda^k)} \overset{\text{def.}}{=} \widetilde{\nu}_\lambda^{k-1}\left(\overline{\theta}_\lambda^k\right).$$

Use the fact that $\underline{\phi}(x)x = 1 - e^{-x}$ combined with the definition of the normalized Newton decrement to simplify both sides of the previous equation. After simplification, we obtain

$$\mathsf{t}(\overline{\theta}_\lambda^k - \theta_\lambda^k) \leq -\log\left(1 - \widetilde{\nu}_\lambda^{k-1}\left(\overline{\theta}_\lambda^k\right)\right).$$

Apply $\underline{\phi}^{-1}$ on both side to have

$$\underline{\phi}^{-1}\left(\mathsf{t}(\overline{\theta}_\lambda^k - \theta_\lambda^k)\right) \leq -\widetilde{\nu}_\lambda^{k-1}\left(\overline{\theta}_\lambda^k\right)^{-1}\log\left(1 - \widetilde{\nu}_\lambda^{k-1}\left(\overline{\theta}_\lambda^k\right)\right),$$

and the conclusion follows with the fact that this is an increasing function of the normalized Newton decrement, which is upper bounded by $\mathsf{c}$. □

This lemma will be useful in the following derivation, and provide some intuition on GSC loss function.

*Remark* 6. *Intuition for GSC loss function.* The purpose of working with Generalized self-concordant loss functions is to be able to control the deviation of the function with their local quadratic approximation. For $\theta \in \mathcal{H}$, Lemma 3 gives us that we can bound quantities depending on $\mathsf{t}$ in the inequalities of GSC loss functions of Proposition 3. When $\theta$ is deep into $\mathsf{D}_\lambda^{k-1}$, then $\mathsf{t} \to 1/2$ and the bounds of Proposition 3 are tight. On the contrary, when $\theta$ leaves this ellipsoid, the upper bound diverges exponentially to infinity while the lower bound goes exponentially to $0$, making the deviation from the quadratic approximation very loose.

To conclude, a GSC function with high $R$ has small Dikin ellipsoids, and is far from its quadratic approximation. On the contrary, a GSC function with low $R$ will be close to its quadratic approximation; the Dikin ellipsoid is large. The extreme case is obtained when $\ell$ is the square loss. Then, $\phi = \{0\}$, so $R = 0$, and the Dikin ellipsoid spans the whole space for any $\theta \in \mathcal{H}$. This implies *e.g.* that the lower and upper bound on the gradient matches, making the quadratic approximation tight.

## C.2 Error propagation

In this section, we give a sufficient condition for achieving an $\epsilon$ error on a sequence of proximal operators. Indeed, we aim at minimizing $L_\lambda^{t-1}$, but we do not have access to this function; only to its approximation $\overline{L}_\lambda^{t-1}$. Relating both is the purpose of the next result.

**Proposition 7 (Error propagation with proximal sequence).** *Let $\mathsf{c} > 0$. Assume that you can solve each subproblem with precision $\overline{\epsilon}_k$ and that you have a guarantee on the exact normalized decrement:*

$$\forall k \in \{1, \ldots, t\}, \quad \begin{cases} \overline{\nu}_\lambda^{k-1}\left(\overline{\theta}_\lambda^k\right) \leq \overline{\epsilon}_k \\ \overline{\theta}_\lambda^k \in \mathsf{D}_\lambda^{k-1}(\mathsf{c}) \iff \widetilde{\nu}_\lambda^{k-1}\left(\overline{\theta}_\lambda^k\right) \leq \mathsf{c} \end{cases}$$

*Then requiring:*

$$\forall k \in \{1, \ldots, t\}, \quad \overline{\epsilon}_k = \epsilon \frac{\kappa_\mathsf{c}^{k-t}}{t}$$

*with $\kappa_\mathsf{c} = -1/\mathsf{c}\log(1 - \mathsf{c})$ suffice to achieve an error $\epsilon$:*

$$\nu_\lambda^{t-1}\left(\overline{\theta}_\lambda^t\right) \leq \epsilon.$$

*We can replace the condition $\overline{\theta}_\lambda^k \in \mathsf{D}_\lambda^{k-1}(\mathsf{c})$ with $\epsilon \leq \mathsf{c}\sqrt{\lambda}/R$.*

*Proof.* Let us track the error step by step. Denote by $\epsilon_k$ the Newton decrement of the exact function at each step:

$$\forall k, \quad \epsilon_k \stackrel{\text{def.}}{=} \nu_\lambda^{k-1}\left(\bar{\theta}_\lambda^k\right).$$

Consider the following decomposition at step $k$:

$$
\begin{aligned}
\nu_\lambda^{k-1}\left(\bar{\theta}_\lambda^k\right) &= \left\|\nabla L(\bar{\theta}_\lambda^k) + \lambda\left(\bar{\theta}_\lambda^k - \theta_\lambda^{k-1}\right)\right\|_{\mathbf{H}_\lambda^{-1}(\bar{\theta}_\lambda^k)} \\
&\leq \left\|\nabla L(\bar{\theta}_\lambda^k) + \lambda\left(\bar{\theta}_\lambda^k - \bar{\theta}_\lambda^{k-1}\right)\right\|_{\mathbf{H}_\lambda^{-1}(\bar{\theta}_\lambda^k)} + \left\|\lambda\left(\bar{\theta}_\lambda^{k-1} - \theta_\lambda^{k-1}\right)\right\|_{\mathbf{H}_\lambda^{-1}(\bar{\theta}_\lambda^k)} \\
&\leq \bar{\nu}_\lambda^{k-1}\left(\bar{\theta}_\lambda^k\right) + \lambda\left\|\mathbf{H}_\lambda^{-1/2}(\bar{\theta}_\lambda^k)\mathbf{H}_\lambda^{-1/2}(\theta_\lambda^k)\right\|\left\|\bar{\theta}_\lambda^{k-1} - \theta_\lambda^{k-1}\right\|_{\mathbf{H}_\lambda(\bar{\theta}_\lambda^{k-1})} \\
&\leq \bar{\nu}_\lambda^{k-1}\left(\bar{\theta}_\lambda^k\right) + \underline{\phi}^{-1}\left(\mathsf{t}(\bar{\theta}_\lambda^{k-1} - \theta_\lambda^{k-1})\right)\nu_\lambda^{k-2}\left(\bar{\theta}_\lambda^{k-1}\right)
\end{aligned}
$$

In the last inequality we used that $\left\|\mathbf{H}_\lambda^{-1/2}(\bar{\theta}_\lambda^k)\mathbf{H}_\lambda^{-1/2}(\theta_\lambda^k)\right\| \leq 1/\lambda$ and the relation between the distance in Hessian's norm and the Newton decrement of Eq. (58). Introducing the notation with epsilon, the last line is by definition

$$\epsilon_k \leq \bar{\epsilon}_k + \underline{\phi}^{-1}\left(\mathsf{t}(\bar{\theta}_\lambda^{k-1} - \theta_\lambda^{k-1})\right)\epsilon_{k-1}. \tag{59}$$

The first term $\bar{\epsilon}_k$ is the error we can control at each step whereas $\epsilon_{k-1}$ is the error of interest which increases with $k$. Using the fact that $\bar{\theta}_\lambda^{k-1} \in \mathsf{D}_\lambda^{k-2}(\mathsf{c})$, we have

$$\underline{\phi}^{-1}\left(\mathsf{t}(\bar{\theta}_\lambda^{k-1} - \theta_\lambda^{k-1})\right) \leq -\frac{1}{\mathsf{c}}\log(1-\mathsf{c}) \stackrel{\text{def.}}{=} \kappa_\mathsf{c}$$

thanks to Lemma 3. Thus, Eq. (59) becomes

$$\epsilon_k \leq \bar{\epsilon}_k + \kappa_\mathsf{c}\epsilon_{k-1}. \tag{60}$$

This being valid for all $i \leq k$ and since $\bar{\nu}_\lambda^0\left(\bar{\theta}_\lambda^1\right) = \nu_\lambda^0\left(\bar{\theta}_\lambda^1\right)$ we obtain that

$$\epsilon_k \leq \sum_{i=1}^k \bar{\epsilon}_i \kappa_\mathsf{c}^{k-i}. \tag{61}$$

Now plug the assumption of the proposition, namely that each problem is solved with precision

$$\bar{\epsilon}_k = \epsilon\frac{\kappa_\mathsf{c}^{k-t}}{t}$$

and use Eq. (61) at step $t$ to obtain

$$\epsilon_t \leq \sum_{i=1}^t \kappa_\mathsf{c}^{i-t}\kappa_\mathsf{c}^{t-i}\frac{\epsilon}{t} = \epsilon. \tag{62}$$

**Replacing $\bar{\theta}_\lambda^k \in \mathsf{D}_\lambda^{k-1}(\mathsf{c})$ with $\epsilon \leq \mathsf{c}\sqrt{\lambda}/R$.** Let $k \geq 1$. Then, having

$$\bar{\theta}_\lambda^k \in \mathsf{D}_\lambda^{k-1}(\mathsf{c})$$

amounts by definition to have

$$\widetilde{\nu}_\lambda^{k-1}\left(\bar{\theta}_\lambda^k\right) \leq \mathsf{c},$$

which is also equivalent to

$$\nu_\lambda^{k-1}\left(\bar{\theta}_\lambda^k\right) \leq \mathsf{c}\,\mathsf{r}_\lambda(\bar{\theta}_\lambda^k).$$

We can use the crude lower bound $\mathsf{r}_\lambda(\cdot) \geq \sqrt{\lambda}/R$. Thus, the following implication holds:

$$\epsilon_k \stackrel{\text{def.}}{=} \nu_\lambda^{k-1}\left(\bar{\theta}_\lambda^k\right) \leq \mathsf{c}\frac{\sqrt{\lambda}}{R} \implies \bar{\theta}_\lambda^k \in \mathsf{D}_\lambda^{k-1}(\mathsf{c}). \tag{63}$$

Now, assume $\epsilon \leq \mathsf{c}\sqrt{\lambda}/R$. Then, we have that

$$\epsilon_1 = \bar{\epsilon}_1 = \epsilon \frac{\kappa_\mathsf{c}^{1-t}}{t} \leq \mathsf{c}\sqrt{\lambda}/R \frac{\kappa_\mathsf{c}^{1-t}}{t},$$

which gives

$$\epsilon_1 \leq \mathsf{c}\sqrt{\lambda}/R,$$

which implies $\overline{\theta}_\lambda^1 \in \mathsf{D}_\lambda^0(\mathsf{c})$ following Eq. (63). Then Eq. (60) holds with $k = 2$:

$$\epsilon_2 \leq \bar{\epsilon}_2 + \kappa_\mathsf{c}\epsilon_1.$$

For bigger $k$, proceed by induction. Let $k < t$ and assume for any $i < k$ that

$$\epsilon_{i+1} \leq \bar{\epsilon}_{i+1} + \kappa_\mathsf{c}\epsilon_i.$$

Then, we have that

$$\epsilon_k \leq \sum_{i=1}^k \bar{\epsilon}_i \kappa_\mathsf{c}^{k-i}$$

which gives the following bound, thanks to the assumption on $\epsilon$ and the $\bar{\epsilon}_i$:

$$\epsilon_k \leq \sum_{i=1}^k \epsilon \frac{\kappa_\mathsf{c}^{i-t}}{t} \kappa_c^{k-i} \leq \mathsf{c}\sqrt{\lambda}/R.$$

This implies $\overline{\theta}_\lambda^k \in \mathsf{D}_\lambda^{k-1}(\mathsf{c})$ following Eq. (63), and Eq. (60) holds at step $k + 1$. Thus the induction hypothesis holds for all $k$ and the conclusion of Eq. (62) holds.

**Proof of Proposition 1.** This result is a direct application of the previous one, where we set $\mathsf{c} = 1/2$. □

We see that the requirement $\epsilon \leq \mathsf{c}\sqrt{\lambda}/R$ is simply to ensures that a bound on the Newton decrement $\nu_\lambda^{k-1}\left(\overline{\theta}_\lambda^k\right)$ translates to a bound on the *normalized* Newton decrement $\widetilde{\nu}_\lambda^{k-1}\left(\overline{\theta}_\lambda^k\right)$ *via* the crude bound on the Dikin radius $\mathsf{r}_\lambda(\overline{\theta}_\lambda^k) \geq R/\sqrt{\lambda}$. Thus, the requirement on $\epsilon$ can be dropped if we assume $\overline{\theta}_\lambda^k \in \mathsf{D}_\lambda^{k-1}(\mathsf{c})$. Such condition is enforced in solver such as the one developed in [14].

Finally, we put in application this result with next proposition, which gives a bound on the excess risk with inexact solver.

**Proposition 8 (Bound on the excess risk with inexact solver).** *Assume that:*

- *the requirement of Proposition 7 hold;*

- *the requirement of Theorem 4 hold, namely* $\mathsf{H}_{1-5}$*;*

*The first is an hypothesis on the* optimization procedure*, while the second in an hypothesis on the* statistics *of the learning task. Then, denoting* $\overline{\theta}_\lambda^t$ *the approximation of* $\widehat{\theta}_\lambda^t$ *as defined in Proposition 7, we have the following bound on the excess risk:*

$$L(\overline{\theta}_\lambda^t) - L(\theta^\star) \leq \mathsf{C}_{\text{bias}}\lambda^{2s} + \mathsf{C}_{\text{var}}\frac{\mathsf{df}_\lambda}{n} + \mathsf{E}_\mathsf{c}\,\epsilon, \quad s = (r + 1/2) \wedge t,$$

*with:*

$$\mathsf{E}_\mathsf{c} \overset{\text{def.}}{=} 4\Psi(4 - \log(1 - \mathsf{c}))\frac{e^4}{1 - \mathsf{c}}\kappa_\mathsf{c}^2, \quad e.g. \quad \mathsf{E}_{1/2} \leq 4.3 \cdot 10^3.$$

*Proof.* The proof boils down to combining the statistical results held in Theorem 4 with the optimization result of Proposition 7. Begin by writing

$$L(\widehat{\theta}_\lambda^t) - L(\theta^\star) \leq \Psi\left(\mathsf{t}(\overline{\theta}_\lambda^t - \theta^\star)\right)\left\|\overline{\theta}_\lambda^t - \theta^\star\right\|_{\mathbf{H}(\theta^\star)}^2$$

$$\leq 2\Psi\left(\mathsf{t}(\widehat{\theta}_\lambda^t - \overline{\theta}_\lambda^t) + \mathsf{t}(\widehat{\theta}_\lambda^t - \theta^\star)\right)\left[\left\|\widehat{\theta}_\lambda^t - \overline{\theta}_\lambda^t\right\|_{\mathbf{H}(\theta^\star)}^2 + \left\|\widehat{\theta}_\lambda^t - \theta^\star\right\|_{\mathbf{H}(\theta^\star)}^2\right].$$

We know how to handle the statistical term $\left\|\widehat{\theta}_\lambda^t - \theta^\star\right\|_{\mathbf{H}(\theta^\star)}^2$.

**Bound on** $\mathsf{t}(\widehat{\theta}_\lambda^t - \overline{\theta}_\lambda^t)$**.** As in the beginning of the proof of Proposition 4, we write:

$$
\begin{aligned}
\mathsf{t}(\widehat{\theta}_\lambda^t - \overline{\theta}_\lambda^t) &\leq \frac{1}{\mathsf{r}_\lambda(\overline{\theta}_\lambda^t)} \left\|\overline{\theta}_\lambda^t - \widehat{\theta}_\lambda^t\right\|_{\mathbf{H}_\lambda(\overline{\theta}_\lambda^t)} \\
&\leq \frac{1}{\mathsf{r}_\lambda(\overline{\theta}_\lambda^t)} \underline{\phi}^{-1}\left(\mathsf{t}(\overline{\theta}_\lambda^t - \widehat{\theta}_\lambda^t)\right) \left\|\nabla\widehat{L}_\lambda^{t-1}(\overline{\theta}_\lambda^t)\right\|_{\mathbf{H}_\lambda^{-1}(\overline{\theta}_\lambda^t)} \\
&= \underline{\phi}^{-1}\left(\mathsf{t}(\overline{\theta}_\lambda^t - \widehat{\theta}_\lambda^t)\right) \widetilde{\nu}_\lambda^{t-1}\left(\overline{\theta}_\lambda^t\right) \\
&\leq \underline{\phi}^{-1}\left(\mathsf{t}(\overline{\theta}_\lambda^t - \widehat{\theta}_\lambda^t)\right) \mathsf{c}
\end{aligned}
$$

where we used the fact that $\overline{\theta}_\lambda^t \in \mathsf{D}_\lambda^{t-1}(\mathsf{c})$, an assumption of Proposition 7. With the same reasoning of Eq. (52), we conclude:

$$
\mathsf{t}(\widehat{\theta}_\lambda^t - \overline{\theta}_\lambda^t) \leq -\log(1 - \mathsf{c}).
$$

**Bound on** $\left\|\widehat{\theta}_\lambda^t - \overline{\theta}_\lambda^t\right\|_{\mathbf{H}(\theta^\star)}^2$**.** Use a similar reasoning as we used for the variance. Under $\mathsf{H}_1$, we have (Proof of Theorem 2, 1st point)

$$
\left\|\widehat{\theta}_\lambda^t - \overline{\theta}_\lambda^t\right\|_{\mathbf{H}(\theta^\star)}^2 \leq 2\left\|\widehat{\theta}_\lambda^t - \overline{\theta}_\lambda^t\right\|_{\widehat{\mathbf{H}}_\lambda(\theta^\star)}^2.
$$

First write:

$$
\left\|\widehat{\theta}_\lambda^t - \overline{\theta}_\lambda^t\right\|_{\widehat{\mathbf{H}}_\lambda(\theta^\star)} \leq e^{\mathsf{t}(\widehat{\theta}_\lambda^t - \theta^\star)/2} e^{\mathsf{t}(\overline{\theta}_\lambda^t - \widehat{\theta}_\lambda^t)/2} \left\|\widehat{\theta}_\lambda^t - \overline{\theta}_\lambda^t\right\|_{\widehat{\mathbf{H}}_\lambda(\overline{\theta}_\lambda^t)},
$$

then, for each term, use:

- $\mathsf{t}(\widehat{\theta}_\lambda^t - \theta^\star) \leq 4$ (end of Theorem 4) so that $e^{\mathsf{t}(\widehat{\theta}_\lambda^t - \theta^\star)/2} \leq e^2$;

- $\mathsf{t}(\overline{\theta}_\lambda^t - \widehat{\theta}_\lambda^t) \leq -\log(1 - \mathsf{c})$ so that $e^{\mathsf{t}(\overline{\theta}_\lambda^t - \widehat{\theta}_\lambda^t)/2} \leq (1 - \mathsf{c})^{-1/2}$;

- and finally:

$$
\begin{aligned}
\left\|\widehat{\theta}_\lambda^t - \overline{\theta}_\lambda^t\right\|_{\widehat{\mathbf{H}}_\lambda(\overline{\theta}_\lambda^t)} &\leq \underline{\phi}^{-1}\left(\mathsf{t}(\overline{\theta}_\lambda^t - \widehat{\theta}_\lambda^t)\right) \left\|\nabla\widehat{L}_\lambda^{t-1}(\overline{\theta}_\lambda^t)\right\|_{\mathbf{H}_\lambda^{-1}(\overline{\theta}_\lambda^t)} \\
&\leq \underline{\phi}^{-1}\left(\mathsf{t}(\overline{\theta}_\lambda^t - \widehat{\theta}_\lambda^t)\right) \nu_\lambda^{t-1}\left(\overline{\theta}_\lambda^t\right) \\
&\leq \left[-\frac{1}{\mathsf{c}}\log(1 - \mathsf{c})\right] \epsilon \stackrel{\text{def.}}{=} \kappa_\mathsf{c}\epsilon.
\end{aligned}
$$

**Putting it all together.** Thus, using the upper bound on the excess risk, we have with probability greater than $1 - 2\delta$

$$
L(\overline{\theta}_\lambda^t) - L(\theta^\star) \leq \mathsf{C}_{\text{bias}}\lambda^{2s} + \mathsf{C}_{\text{var}}\frac{\mathsf{df}_\lambda}{n} + 4\Psi(4 - \log(1 - \mathsf{c}))\frac{e^4}{1 - \mathsf{c}}\kappa_\mathsf{c}^2\epsilon, \quad s = (r + 1/2) \wedge t.
$$

Taking $\mathsf{c} = 1/2$, we have $\Psi(4 - \log(1 - \mathsf{c})) \leq 5$ and $\kappa_\mathsf{c} \leq 1.4$, which allows bounding the quantity in front of $\epsilon$. $\qquad\square$

# D Technical lemmas

## D.1 Concentration of Hermitian operators

In this section, we import results from [1] and [17]. The former provides a bound on $\left\|\widehat{\mathbf{H}}_\lambda^{-1/2}(\theta)\mathbf{H}_\lambda^{1/2}(\theta)\right\|$. The latter provides a bound on $\left\|\widehat{\mathbf{H}}_\lambda^{-1}(\theta)\mathbf{H}_\lambda(\theta)\right\|$, which is more difficult to obtain. They use the fact that $\mathrm{df}_\lambda = \mathrm{Tr}\,\mathbf{H}_\lambda(\theta)\mathbf{H}(\theta)$ for least square, but we can't use this very convenient relation here. Thus, we only use their result in the case $1/2 < r < 1$, which makes optimal rate still possible.

We will only use

$$\mathrm{Tr}\,\mathbf{H}_\lambda^{-1}(\theta)\widehat{\mathbf{H}}(\theta) \le \frac{\mathsf{B}_2(\theta)}{\lambda}.$$

**Proposition 9 (Concentration bound).** *Let $\delta \in (0,1]$ and $\lambda > 0$. The following holds:*

$$n \ge 24\frac{\mathsf{B}_2(\theta)}{\lambda}\log\frac{8\mathsf{B}_2(\theta)}{\lambda\delta} \quad\implies\quad \left\|\widehat{\mathbf{H}}_\lambda^{-1/2}(\theta)\mathbf{H}_\lambda^{1/2}(\theta)\right\| \le \sqrt{2}, \tag{64}$$

$$n \ge 8\frac{\mathsf{B}_2(\theta)^2}{\lambda^2}\log^2\frac{2}{\delta} \quad\implies\quad \left\|\widehat{\mathbf{H}}_\lambda^{-1}(\theta)\mathbf{H}_\lambda(\theta)\right\| \le 2, \tag{65}$$

$$n \ge 2\left(1 \vee \frac{4\mathsf{B}_2(\theta)^2}{\lambda^{2s}}\right)\log\frac{2}{\delta} \quad\implies\quad \left\|\mathbf{H}(\theta)-\widehat{\mathbf{H}}(\theta)\right\|_{HS} \le \lambda^s, \tag{66}$$

*where each bound hold with confidence $1-\delta$.*

*Proof.* The first equation is Lemma 6 of [1]. The second equation can be adapted from Proposition 5.4 of [17], except that we use

$$\mathrm{Tr}\,\mathbf{H}_\lambda(\theta)\mathbf{H}(\theta) \le \frac{\mathsf{B}_2(\theta)}{\lambda}$$

instead of $\mathrm{df}_\lambda$. For the last inequality, use Bernstein inequality for random vectors. With probability $1-\delta$:

$$\left\|\mathbf{H}(\theta)-\widehat{\mathbf{H}}(\theta)\right\|_{HS} \le \frac{2\mathsf{B}_2(\theta)\log 2/\delta}{n} + \mathsf{B}_2(\theta)\sqrt{\frac{2\log 2/\delta}{n}}.$$

Assuming $n \ge 2\log 2/\delta$, this bound becomes

$$\left\|\mathbf{H}(\theta)-\widehat{\mathbf{H}}(\theta)\right\|_{HS} \le 2\mathsf{B}_2(\theta)\sqrt{\frac{2\log 2/\delta}{n}}.$$

Let $s > 0$. Further requiring $n \ge 8\mathsf{B}_2(\theta)\lambda^{-2s}\log 2/\delta$ gives:

$$\left\|\mathbf{H}(\theta)-\widehat{\mathbf{H}}(\theta)\right\|_{HS} \le \lambda^s$$

which completes the proof. $\qquad\square$

## D.2 Inequalities on Hermitian operators

The following results are given in [17]. We redo the proof to track down and upper bound the constants which are discarded in the original paper.

**Lemma 4 (Hermitian operator inequalities).** *Let $A, B$ be two non-negative self-adjoint operators on $\mathcal{H}$. Assume $\|A\|, \|B\| \le \kappa$, where $\|\cdot\|$ denotes the operator norm. Then:*

$$\forall r \le 1, \quad \|A^r - B^r\| \le \|A-B\|^r \tag{67}$$

$$\forall r > 1, \quad \|A^r - B^r\| \le w(r)\|A-B\| \tag{68}$$

$$\forall r \le 1, \quad \|A^r B^r\| \le \|AB\|^r \tag{69}$$

*with $r2^{\lfloor r\rfloor+1}\kappa^r$.*

*Proof.* For the first point, refer to [29] Theorem X.1.I, Eq. (X.2). For the third point, refer to Theorem IX.2.1 of the same book. It is also known as Cordes inequality [30]. The proofs involve positive semidefinite matrices but are directly applicable to non-negative self-adjoint Hermitian operators.

For the second point, assume $\|A\|, \|B\| \leq 1$. Consider the function $f(x) = (1 - x)^r$, defined for $|x| \leq 1$. Its Taylor expansion reads:

$$f(x) = \sum_{n \geq 0} a_n x^n, \quad a_n = \frac{(-1)^n}{n!} \prod_{k=1}^{n} (r - k + 1)$$

We have:

$$\left| \frac{a_{n+1} x^{n+1}}{a_n x^n} \right| = \left| \frac{r - n}{n + 1} \cdot x \right| \underset{n \to \infty}{\to} |x|$$

so applying d'Alembert's rule, we have that the radius of the serie is 1. Now, we have that:

$$A^r - B^r = f(\mathbf{I} - A) - f(\mathbf{I} - B) = \sum_{n \geq 0} a_n \left[ (\mathbf{I} - A)^n - (\mathbf{I} - B)^n \right]$$

$$\implies \|A^r - B^r\| \leq \sum_{n \geq 0} |a_n| \, \|(\mathbf{I} - A)^n - (\mathbf{I} - B)^n\|$$

Using that $(\mathbf{I} - A)^n - (\mathbf{I} - B)^n = (\mathbf{I} - A)(\mathbf{I} - A)^{n-1} - (\mathbf{I} - B)^{n-1} - (B - A)(\mathbf{I} - B)^{n-1}$, we obtain:

$$\|(\mathbf{I} - A)^n - (\mathbf{I} - B)^n\| \leq \left\| (\mathbf{I} - A)(\mathbf{I} - A)^{n-1} - (\mathbf{I} - B)^{n-1} \right\| + \left\| (B - A)(\mathbf{I} - B)^{n-1} \right\|$$
$$\leq \left\| (\mathbf{I} - A)^{n-1} - (\mathbf{I} - B)^{n-1} \right\| + 1$$
$$\leq n \, \|A - B\|$$

Denoting $g(x) = (1 - x)^{r-1} = \sum b_n x^n$, we have $f'(x) = -rg(x)$ which gives $n \, |a_n| = r \, |b_n|$. Then:

$$\|A^r - B^r\| \leq \|A - B\| \sum_{n \geq 0} n \, |a_n|$$

$$\leq r \, \|A - B\| \sum_{n \geq 0} |b_n|$$

We can somewhat painfully upper bound this last term. Notice that for $n > r$, all the $b_n$ have the same sign $s = (-1)^{\lfloor r \rfloor}$. Thus, for $N > r$:

$$\sum_{n=0}^{N} |b_n| = \sum_{n=0}^{\lfloor r \rfloor} |b_n| + s \sum_{n=\lfloor r \rfloor}^{N} b_n$$

$$= \sum_{n=0}^{\lfloor r \rfloor} |b_n| + s \lim_{x \to 1} \sum_{n=\lfloor r \rfloor}^{N} b_n x^n$$

$$\leq 2 \sum_{n=0}^{\lfloor r \rfloor} |b_n| + \lim_{x \to 1} g(x)$$

$$\leq 2 \sum_{n=0}^{\lfloor r \rfloor} \frac{1}{n!} \prod_{k=1}^{n} (r - k + 1)$$

$$\leq 2 \sum_{n=0}^{\lfloor r \rfloor} \binom{\lfloor r \rfloor}{n} = 2^{\lfloor r \rfloor + 1}$$

Finally, apply these properties to $A/\kappa, B/\kappa$ to obtain in general:

$$\|A^r - B^r\| \leq r 2^{\lfloor r \rfloor + 1} \kappa^r \, \|A - B\|$$

$\square$

### D.3 Basic calculus

This is a few line of computation, but useful in multiple places.

**Lemma 5 (Bound on residual of IT's spectral function).** *Let $r, t > 0$. Consider the following function defined on $[0, \kappa]$:*

$$h(\sigma) = \left( \frac{\lambda}{\lambda + \sigma} \right)^t \sigma^r.$$

*Then:*

$$\sup_{0 \le \sigma \le \kappa} h(\sigma) \le \begin{cases} \left( r \cdot \frac{\lambda}{t} \right)^r & \text{if } r < t \\ \left( \frac{\lambda}{\kappa + \lambda} \right)^t \kappa^r & \text{otherwise}. \end{cases}$$

*Proof.* $h$ is differentiable and

$$h'(\sigma) = \frac{\lambda^t \sigma^{r-1}}{(\sigma + \lambda)^{t+1}} \left[ \sigma(r - t) + r\lambda \right].$$

If $t \le r$, the regularization saturates and the maximum is in $\hat{\sigma} = \kappa$, which gives

$$\sup_{\sigma} h(\sigma) \le \left( \frac{\lambda}{\kappa + \lambda} \right)^t \kappa^r \underset{\lambda \to 0}{\sim} \lambda^t \kappa^{r-t}.$$

Otherwise, if $t > r$, the maximum is in $\hat{\sigma} = \frac{r\lambda}{t-r}$ and it reads

$$\begin{aligned} \sup_{\sigma} h(\sigma) &\le \left( \frac{t - r}{t} \right)^t \left( \frac{r\lambda}{t - r} \right)^r \\ &= \left( \frac{t - r}{t} \right)^{t-r} r^r \left( \frac{\lambda}{t} \right)^r. \end{aligned} \tag{70}$$

We can rewrite the prefactor in front of $(\lambda/t)^r$. First,

$$\left( \frac{t - r}{t} \right)^{t-r} r^r = \left( \frac{t - r}{t} \right)^t \left( \frac{rt}{t - r} \right)^r.$$

Then, use

$$\left( \frac{t - r}{t} \right)^t \le e^{-r} \quad \text{when} \quad r < t. \tag{71}$$

Also,

$$\left( \frac{rt}{e(t - r)} \right)^r = \left( e \left( \frac{1}{r} - \frac{1}{t} \right) \right)^{-r} \le (e/r)^{-r} \le r^r. \tag{72}$$

Use Eq. (71) and Eq. (72) on the upper bound of Eq. (70), and the result is obtained. $\square$

## E Experiments

### E.1 Technical details

**Splines.** The spline kernel of order $q$ is defined on $[0, 1]^2$ as

$$\Lambda_q(x, z) = \sum_{k \in \mathbb{Z}} \frac{e^{2i\pi k(x-z)}}{|k|^q}.$$

A closed form expression is available when $q$ is an even integer:

$$\Lambda_q(x, z) = 1 + \frac{(-1)^{q/2-1}}{q!} B_q(|x - z|).$$

$B_q$ are Bernoulli polynomial of order $q$. They can be implemented easily. We also have the relation

$$\langle \Lambda_q(x, \cdot), \Lambda_{q'}(x', \cdot) \rangle_{L_2(\mathcal{X}, \rho_\mathbf{x})} = \Lambda_{q+q'}(x, x')$$

Our choice of $r, \alpha$ reflects the constraints on $\alpha$ and $(r + 1/2)\alpha + 1/2$ to be even integers.

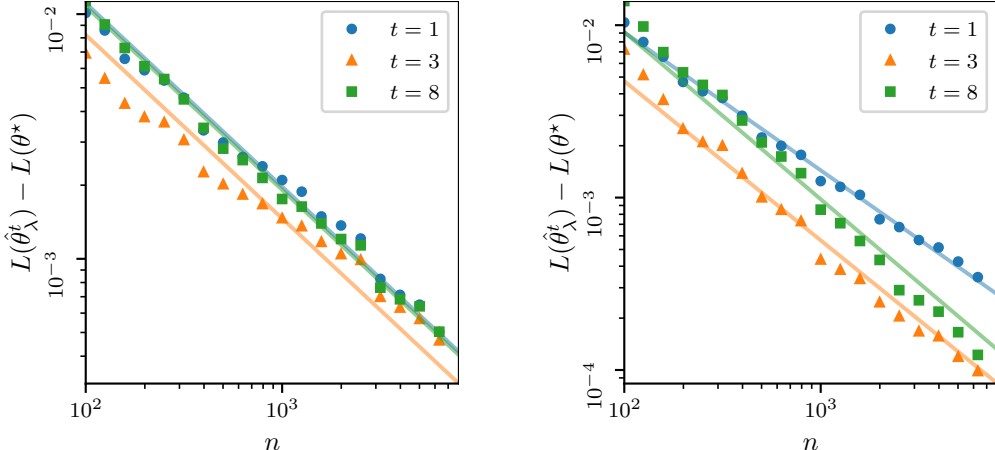

Figure 2: Excess risk with **least square** for various Iterated Tikhonov estimator, function of $n$. **Colors**: $t = 1$ (Tikhonov) estimator is shown in orange; $t = 2, 3$ in green, red. **Left**: from a difficult problem, $r = 1/4, \alpha = 2$. **Right**: easy problem, $r = 41/4, \alpha = 2$. Plain lines are predicted by theory, with slope $-\alpha(1+2s)/1+\alpha(1+2s)$, $s = \min\{r, t - 1/2\}$ (see main text). All plots are averaged over 100 different initialization.

**Regularization.** For both least square and logistic regression, the regularization $\lambda$ is chosen among $50$ log spaced values between $10^{-4}$ and $1$.

**Resources.** Computation was carried by a `Intel(R) Xeon(R) CPU E5-1620 v2 @ 3.70GHz`, with 32GB of RAM.

## E.2 Simulations with least square

**Estimating $\widehat{\theta}_\lambda^t$.** We leverage the very convenient filter interpretation with least square. We diagonalize the kernel matrix $K = UDU^\top$ once, then evaluate the estimator with

$$\widehat{\theta}_\lambda^t = \sum_{i=1}^{n} \alpha_i \phi(x_i),$$

$$\alpha = \frac{1}{n} U g_\lambda^t (D/n) D^\top y,$$

where $g_\lambda^t$ is IT's filter, defined in (8).

**Simulations.** The simulations are reported in Figs. 2 and 3. The same broad conclusion as for the classification task with the logistic loss apply. Surprisingly, IT (8) seems to suffer from higher constant than its counterpart with low $t$.

## E.3 Synthetic binary task

**Derivation of the noise.** We have $\theta^\star(x) = \Lambda_{(r+1/2)\alpha+\epsilon}(x, 0)$ a function of smoothness $r + 1/2$ in $L_2(\mathcal{X}, \rho_\mathbf{x})$. We want to use logistic regression. Thus, we need to choose the noise $\rho(y \mid x)$ so that

$$\theta^\star(x) = \arg\min_z \int_\mathcal{Y} \ell(y, z) \mathrm{d}\rho_{y|x}(y).$$

To keep things simple, we restrict the output space to $\mathcal{Y} = \{-1, 1\}$. Denote $a(x) = \mathbb{P}(y = 1 \mid x)$. We will have $\mathbb{P}(y = -1 \mid x) = 1 - a(x)$. Now we need to choose $a$ s.t

$$a \in [0, 1] \quad \text{and} \quad \theta^\star(x) = \arg\min_z h(z) \overset{\text{def.}}{=} \log(1 + e^z)(1 - a) + \log(1 + e^{-z})a.$$

Having $a > 0, 1 - a > 0$ implies that $h$ has a unique minimizer $z^*$. Then

$$h'(z) = \frac{1}{1 + e^z}\left((1 - a)e^z - a\right) \implies a = \frac{e^{z^*}}{1 + e^{z^*}}.$$

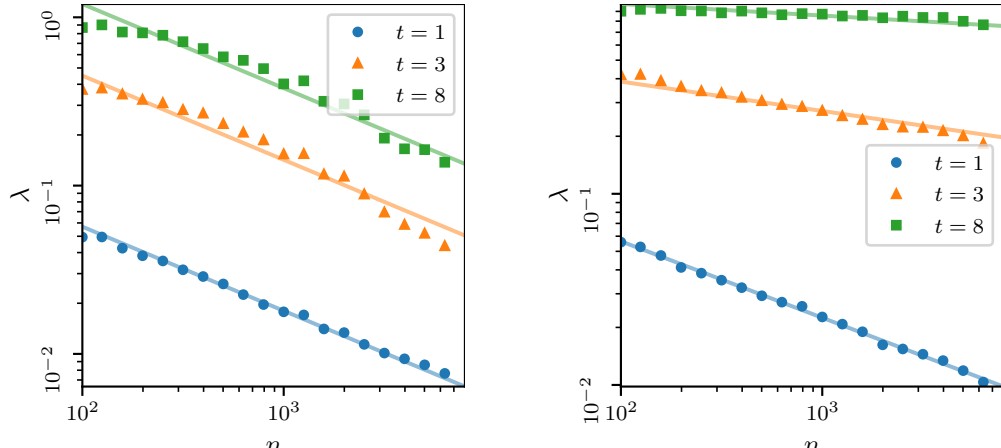

Figure 3: Chosen regularization $\lambda$ with **least square** for various Iterated Tikhonov estimator, function of $n$. **Colors**: $t = 1$ (Tikhonov) estimator is shown in orange; $t = 3, 8$ in green, red. **Left**: from a difficult problem, $r = 1/4, \alpha = 2$. **Right**: easy problem, $r = 41/4, \alpha = 2$. Plain lines are predicted by theory, with slope $-\alpha/1+\alpha(1+2s)$, $s = \min\{r, t - 1/2\}$ (see main text). All plots are averaged over 100 different initialization.

Having required that $\theta^\star(x) = \arg\min_z h(z) \stackrel{\text{def.}}{=} z^*$, we can use the following output distribution:

$$\mathcal{Y} = \{-1, 1\}$$

$$\mathbb{P}(y = 1 \mid x) = \frac{1}{1 + e^{-\theta^\star(x)}}$$

$$\mathbb{P}(y = -1 \mid x) = \frac{1}{1 + e^{+\theta^\star(x)}}$$

which, in turn, ensures that $a(x) \in [0, 1]$.

**Newton or first-order methods.** In practice, the proximal operator is evaluated with a Newton method, or we use the toolbox Cyanure for big $n$ [31]. Both are used with tolerance $10^{-10}$, that is machine precision for single precision. Generally speaking, first-order methods are considered more performant than Newton methods. However, both practical and theoretical considerations motivate the use of second-order scheme in our statement of Proposition 1. Firstly, preconditionated iterative solver such as the one used in [14] provide very efficient results for ill-conditioned problems. Secondly, the analysis of GSC loss functions is well-suited to second-order scheme, as the Newton decrement is a natural quantity to keep track of the optimization error. Measuring the error differently would require additional assumption on the loss function.

**Estimating the excess risk.** The excess risk is estimated with Monte Carlo sampling, with $10^4$ points:

$$ER(\theta) - ER(\theta^\star) \approx \frac{1}{n_{MC}} \sum_{i=1}^{n_{MC}} \frac{1}{1 + e^{-\theta^\star(x_i)}} \log\left(\frac{1 + e^{-\theta(x_i)}}{1 + e^{-\theta^\star(x_i)}}\right) + \frac{1}{1 + e^{\theta^\star(x_i)}} \log\left(\frac{1 + e^{\theta(x_i)}}{1 + e^{\theta^\star(x_i)}}\right)$$

**Additional results.** We report here the regularization $\lambda$ chosen function of $n$ and $t$ for various IT regularized estimators. We confirm that the penalty used for IT is larger than of Tikhonov, to compensate for the fitting induced by the additional proximal steps. We also compare the excess risk achieved by IT with the excess risk of Tikhonov, and observe consistent improvement for easy task with a sufficiently high number of samples.

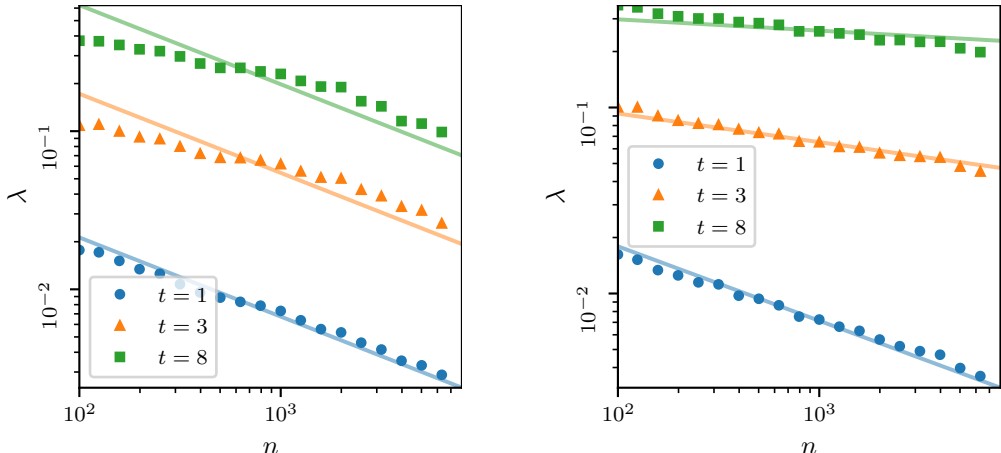

Figure 4: Chosen regularization $\lambda$ for various Iterated Tikhonov estimator, function of $n$. **Colors**: $t = 1$ (Tikhonov) estimator is shown in orange; $t = 3, 8$ in green, red. **Left**: from a difficult problem, $r = 1/4, \alpha = 2$. **Right**: easy problem, $r = 41/4, \alpha = 2$. Plain lines are predicted by theory, with slope $-\alpha/1 + \alpha(1+2s)$, $s = \min\{r, t - 1/2\}$ (see main text). All plots are averaged over 100 different initialization.

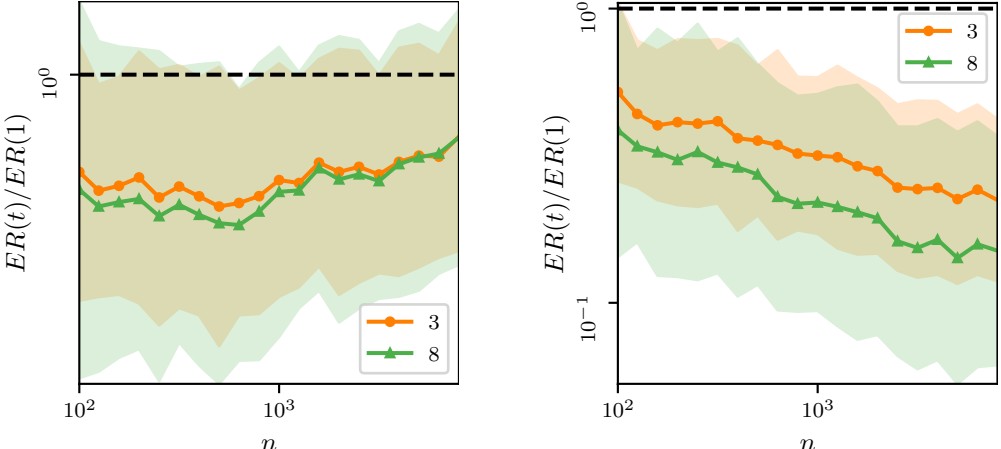

Figure 5: Ratio of IT's excess risk over Tikhonov's excess risk, function of $n$. **Left**: from a difficult problem, $r = 1/4, \alpha = 2$. **Right**: easy problem, $r = 41/4, \alpha = 2$. Whereas we expect the ratio to be consistently lower than 1, IT performs worse than Tikhonov in isolated cases, probably due to the optimization process and the chosen regularization path. Yet, it provides lower excess risk than Tikhonov overall, with up to an order of magnitude of improvement with as few as 1000 samples. All plots are averaged over 100 different initialization.