# OpenReview forum: "Beyond Tikhonov: faster learning with self-concordant losses, via iterative regularization"
_NeurIPS.cc/2021/Conference — NeurIPS 2021 Spotlight_

### Official Review · Reviewer_CRtT · 2021-07-16

**Rating:** 7
**Confidence:** 4

**Summary:**

In this paper, the authors study a supervised learning method called iterative Tikhonov (IT) regularization where the loss function is a generalized self-concordant function. The method is a proximal point algorithm:
- minimize (empirical loss) + (squared Euclidean distance from the previous iterate),

run for t iterations. The authors show that their method (IT) achieves optimal excess risk in interesting regimes of signal smoothness (r) and difficulty of the loss function (\alpha).

They show the benefit of running multiple iterations of the proximal point algorithm when the signal is smooth. They also analyze their method when the proximal step has some optimization error.


**Limitations And Societal Impact:**

Yes

**Main Review:**

The authors discuss relevant works appropriately. They explain their definitions and assumptions. The manuscript is well-structured.

I have a few questions and comments.

1. What does Theorem 1 yield as t goes to \infty? As t goes to Infinity we should get the unregularized estimator, but the theorem gives good rates. In the statement of Theorem 1, please explicitly point to the assumptions you are making.

2. Line 235, why is the first term called bias?

3. In Assumption 9 in the supplement, there is a lower bound on df_\lambda, but it is not present in the main paper. Can you please address the discrepancy, for my understanding?

Minor:
- Line 122: \theta* is the true signal, but here it is called the estimator
- Explain the [h,h] and [k,h,h] notation in the self-concordance condition
- Regarding Definition 4, for Tikhonov regularization, isn't q=1, not just ½? What are $\omega_\nu$ and $\kappa$?
- In Assumption 2, in the end, it should be $\theta,$ not $\theta^*$
- Line 226: should there be a 1/n in front of the summation?
- Line 282: self-reference to Prop. 2
- After line 299: y should be z


**Time Spent Reviewing:**

10

---

> ### Author Response · Authors · 2021-08-10
> **Response to Reviewer CRtT**
>
> We thank this reviewer for their encouraging feedback and for raising interesting questions, as well as for pointing out a few typos (which we will correct).
>
> 1. As observed by the reviewer, when $t$ goes to infinity, we obtain the unregularized estimator. However, there is no paradox with the fact that we achieve the optimal rate. Indeed the leading multiplicative constant of the rate $C_{\\mathrm{var}}$ of Eq. (9) depends linearly on $t$, as shown in Eq. (55) in the Appendix. So, when $t = O(r)$ we achieve the optimal rate and $\mathsf{C}_{\mathrm{var}} = O(r)$. If we let $t$ go to infinity the right hand side diverges. We thank the reviewer for this observation. We will add this comment after Theorem 1.
> 2. We refer to this term as "bias" as it goes to $0$ when the regularization $\lambda$ goes to $0$. Similarly, we refer to the other as the variance as it decays with the number of samples $n$, but increases when $\lambda \to 0$: the degrees of freedom is an increasing function of $\lambda$.
> 3. This is a typo which will be corrected. Thanks for spotting this.
>
> We address the minor comments:
> * $A [k, h, h]$ means "operator $A$ applied to the vectors $k, h, h$.". We will detail this in the definition.
> * Regarding the qualification of Tikhonov regularization, the value of the saturation depends on how the source condition is defined in the filter literature.  In particular, the source condition parameter $r$ defined in terms of the integral operator in $L^2$ corresponds to $r = s + 1/2$, where s is the parameter of the source condition expressed with respect to the covariance operator in $\mathcal{H}$. This is explained in detail in [a] (Eq. 9, and 10). Now, Thikhonov saturates at $1$ in the formulation with the integral operator. This corresponds to $1/2$ in the formulation with the covariance operator, that is the one we use here in the paper. In particular, in this paper, by taking just one single step of Iterated Tikhonov, we recover the saturation of Tikhonov in Thm. 1, which was unveiled by [b].
> * Regarding definition 4, the quantities $\omega_\nu$ and $\kappa$ come from the definition of the qualification given in [a,c], and will be detailed. $\omega_\nu$ is a constant independent of $\lambda$, and $\kappa$ is a bound on the kernel function (e.g. $1$ for the Gaussian kernel).
> * Thanks for spotting all the typos in the last 3 comments. These will be corrected.
>
>
> ### References
> [a] F. Bauer, S. Pereverzev, and L. Rosasco. On regularization algorithms in learning theory.
> Journal of complexity, 23(1):52–72, 2007.
>
> [b] U. Marteau-Ferey, D. Ostrovskii, F. Bach, and A. Rudi. Beyond least-squares: Fast rates for regularized empirical risk minimization through self-concordance. In Conference on Learning Theory (COLT), 2019.
>
> [c] L. Lo Gerfo, L. Rosasco, F. Odone, E. De Vito, and A. Verri. Spectral algorithms for supervised learning. Neural Computation, 20(7):1873–1897, 2008.

---

### Official Review · Reviewer_xAhK · 2021-07-17

**Rating:** 7
**Confidence:** 3

**Summary:**

The authors consider supervised learning problems based on generalized self-concordant (GSC) loss functions and iterated Tikhonov regularization scheme. Unlike the least-squares loss, there is no closed-form expression of the estimator induced by the GSC loss, which makes the generalization error analysis much more complicated. This paper shows that the proposed algorithm can overcome the limitation for GSC functions and achieve optimal learning rates.

**Limitations And Societal Impact:**

Yes.

**Main Review:**

The iterative regularization has a higher qualification compared to the conventional Tikhonov regularization scheme, which can overcome the so-called saturation effect. An important and interesting question is to investigate supervised learning problems associated with iterated Tikhonov regularization and generalized self-concordant (GSC) loss functions which are three times differentiable with their third derivative being bounded by the second derivative (e.g. logistic loss). In this paper, the authors investigate the benefit of iterated Tikhonov regularization in learning algorithms associated with GSC loss and show that the proposed method can still achieve optimal learning rates even when the index of the source condition is large. Sufficient conditions are also given for the computation of the proposed estimator in practice. Overall, the paper is well-written and technically sound, which provides new insights for the error analysis of estimators without closed-form expression. I was wondering if the boundedness of $\theta^*$ is required in the proof. In binary classification, the minimizer of the expected risk associated with logistic loss is usually unbounded (that is, $\theta^*=\ln\frac{\eta}{1-\eta}$ with $\eta(x)=P(Y=1|X=x$)).

**Time Spent Reviewing:**

3

---

> ### Author Response · Authors · 2021-08-10
> **Response to Reviewer xAhK**
>
> We thank the reviewer for their encouraging feedback. Indeed, we hope this paper will pave the way for more results on the statistical properties of estimators without a closed-form expression, notably when using different regularization schemes with GSC loss functions.
> Concerning the question of the reviewer, we agree that it would be interesting to consider unbounded $\theta^\star$. However, an important assumption of our work is that $\theta^\star$ exists and belongs to a reproducing kernel Hilbert space $\mathcal{H}$, which makes it typically uniformly bounded.  Consider possibly unbounded $\theta^\star$ is an interesting extension that would be possible by casting the whole analysis in $L^2$. To do so, however, the functional analytical tools to deal with GSC functions in Hilbert space are not usable anymore and we need to develop new tools for the $L^2$ case. This is a very interesting direction that we leave for future work.

---

### Official Review · Reviewer_rN14 · 2021-07-17

**Rating:** 7
**Confidence:** 3

**Summary:**

In this paper the authors study methods for minimizing excess expected risk. More specifically, they show how to use an iterated Tikhonov regularization scheme, which is well-known for least squares, for losses which are generalized self concordant (GSC). By doing so, they go beyond some of the limitations of the standard Tikhonov regularization.

**Limitations And Societal Impact:**

Yes, they state that the contribution is purely theoretical.

**Main Review:**

Overall the authors present an interesting method for minimizing expected error for generalized self concordant losses. My main concern is regarding differentiating this work from Marteau-Ferey et al. [1]. This deserves an extended discussion to make more apparent the novelty of the current works' contributions, including a more in-depth overview of the difference between the previously-used Tikhonov regularization, and the newly introduced iterated Tikhonov regularization scheme, as the improvements seem to be somewhat incremental. For example, it would help to include an overview of the specific results achieved by Marteau-Ferey et al. to more directly contrast with the new guarantees in this paper.

======Comments======

Generalized self concordance is defined differently in [2] (there, the generalization comes from varying the exponent on the second-derivative term). It would therefore be helpful to provide a comment distinguishing between the two notions of generalized self concordance.

The authors should be made aware of, and include a reference to, Carmon et al. [3], which uses a ball oracle (related to the prox method) for optimizing quasi self concordant losses (a special case of generalized self concordance losses).

"various set of" -> "various sets of"

"other examples when" -> "other examples include when"

"obtain fast learning rate" -> "obtain fast learning rates"

"While Thikonov" -> "While Tikhonov"


[1] U. Marteau-Ferey, D. Ostrovskii, F. Bach, and A. Rudi. Beyond least-squares: Fast rates for regularized empirical risk minimization through self-concordance. In Conference on Learning Theory (COLT), 2019.

[2] T. Sun and Q. Tran-Dinh. "Generalized self-concordant functions: a recipe for Newton-type methods." Mathematical Programming 178, no. 1 (2019): 145-213.

[3] Y. Carmon, A. Jambulapati, Q. Jiang, Y. Jin, Y.T. Lee, A. Sidford, and K. Tian. "Acceleration with a Ball Optimization Oracle." Advances in Neural Information Processing Systems 33 (2020).

=======================================

After reading the authors' response, I appreciate the added clarity in differentiating from Marteau-Ferey et al. [1], and so I have adjusted my score accordingly.

**Time Spent Reviewing:**

4

---

> ### Author Response · Authors · 2021-08-10
> **Response to Reviewer rN14**
>
> We thank the reviewer for pointing out that we must clarify the improvement of our work with respect to [1]. We are happy to provide the following clarification: The algorithm proposed in [1] is significantly slower than the one that we analyze here (in terms of statistical learning rate). Even without doing assumptions on the capacity condition of the problem (i.e. putting $\alpha = 1$), the algorithm in [1] has a learning error that goes to zero in the best case as
> $$  O(  \max [ n^{-2/3}, n^{-(2r+1)/(2r+2)} ]  )  $$
> while our goes to zero as
> $$ O(n^{-(2r+1)/(2r+2)}) $$
> where $r$ is a measure of the difficulty of the learning problem and can be arbitrarily large. In particular, the learning error of their algorithm stops at $O(n^{-2/3})$, while our algorithm may achieve $O(n^{-1})$. We will add such a comparison at the end of the introduction.
>
> Besides, these two work differ in the following manner:
> * The common part is the nature of the loss function, which is GSC, but the regularization scheme is significantly different (Iterated Tikhonov (IT) vs. Tikhonov). Importantly, we show that IT has higher qualification than Tikhonov. In practice, this implies that as soon as the source condition is larger than $1/2$ (which corresponds to an “easy” learning task), IT can have lower generalization error than Tikhonov. Note that the proof to obtain Thm. 1 is significantly different from the proofs in [1].
> * While easy to ignore, the bias from optimization is studied and we give bounds on the precision required to have accurate estimators in practice (Prop. 1).
>
>
> We also thank the reviewer for the relevant references. We will include the reference [2] in the discussion after the definition of GSC functions. Due to lack of space, we did not include many references to the optimization litterature, as our work focuses on the statistical aspect of the proximal point sequence. We will include reference [3] in our discussion (l. 164).

---

### Official Review · Reviewer_Kz6Q · 2021-08-04

**Rating:** 8
**Confidence:** 4

**Summary:**

The paper derives a probabilistic convergence rate of the estimator obtained by the iterative Tikhonov regularization scheme when the hypothesis space is a Hilbert space within which lies the target function.
The analysis relies on the specific type of generalized self concordant losses, as well as some source and capacity conditions.
Through links with the theory of spectral filtering, optimal rates are given when the estimator does not saturate, showing that the iterative regularization scheme provides benefits for a certain class of problems.
Due to the iterative nature of the regularization scheme, an analysis of the error propagation at each iteration is also included, allowing to solve each problem with less and less precision while keeping statistical guarantees.
Numerical experiments on a toy dataset complete this study, showing the accurateness of the predictions.


**Main Review:**

-Originality:
The contribution is original, as it extends previous results for Tikhonov regularization and GSC losses to iterative Tikhonov.
Related work is appropriately cited, and the contributions are clearly stated.

-Quality:
The submission is of high quality. It is technically sound, claims are supported by appropriate mathematical proofs as well as an illustrative numerical study.

Q1: The assumption on the label space Y is very weak (Borel), how could the dimensionality of Y affect the convergence rate of the learning process ?

Q2: The assumption of the target function belonging to the Hilbert space is crucial for the proof as it relies on the geometry of that space, could it be weakened (similarly to the kernel ridge regression case where the target function can be anywhere from the RKHS to L2) to only keep a modified source condition that would characterize alone the smoothness of the target function ?

Q3: In practice, the number of iterations to perform is unknown as we do not have access to the true source condition, how could one set it given a specific dataset ?

-Clarity:
The submission is very well written. A special effort has been made to make the bounds understandable and relatable to previous works, which makes the reading enjoyable.

-Significance
Iterative regularization is an active topic in statistical learning, and the paper offers novel theoretical insights associated to this scheme, representing a significant contribution for the area.

**Time Spent Reviewing:**

8

---

> ### Author Response · Authors · 2021-08-10
> **Response to Reviewer Kz6Q**
>
> We thank the reviewer for their encouraging feedback. We answer the questions below.
>
> Q1: The difficulty of the learning task is controlled by the source condition and the dimensionality of $Y$ enters in a complicated way, which depends on how we build the space $\mathcal{H}$. Typically, in the case of vector-valued kernel regression with squared loss, if $Y = R^T$ with $T < \infty$, and $\mathcal{H} = \mathcal{H}_0 \otimes \mathbb{R}^T$, a loose upper bound of the norm of $v$, in the source condition, shows that it has a dependence of $O(T^{1/2})$. We expect that here the situation could be similar. However, the question is very interesting and deserves additional study, possibly leveraging results from vector-valued reproducing kernel Hilbert spaces.
>
> Q2: This is a very important question. In our proof strategy, all the quantities are vectors or operators of the Hilbert space. This makes the analysis simpler when $\theta^\star$ is in $\mathcal{H}$.  Clearly, if $\theta^\star$ is not in $\mathcal{H}$, we cannot use the same tools, and we need instead to express all these quantities as vectors or operators in $L^2$. In particular, we would also need to find an equivalent of the integral operator for GSC loss functions, which would constitute a very interesting future work.
>
> Q3: This is a good practical point, and we will add this discussion in section 3.2 after Prop. 2.
> Typically the idea is to consider the number of iterations as a hyperparameter, which could be chosen by cross-validation. Then, we would run the algorithm and test the resulting error on a validation set for each iteration. We will keep doing proximal steps as long as the validation loss improves.

---

> > ### Comment · Reviewer_Kz6Q · 2021-09-10
> > **Response to authors**
> >
> > Thank you for your detailed answer.
> >
> > I will keep my score as it is, and look forward to reading future work further developing the aforementioned perspectives.

---

### Decision · Program_Chairs · 2021-09-28

**Decision:**

Accept (Spotlight)

**Comment:**

The paper analyzes an iterative L2/Tikhonov regularization scheme for statistical learning with general self-concordant losses. All reviewers agree that the paper is technically quite interesting and well-executed, and I believe it will make a nice contribution to the conference program.

Please add the discussion comparing to Marteau-Ferey et al., to the final version, as this was quite helpful for clearing up some confusions in the discussion phase.

**Consistency Experiment:**

NeurIPS has a long history of experimentation. In 2014, NeurIPS ran an experiment in which 10% of submissions were reviewed by two independent committees to quantify the randomness in the review process. This year, we repeated a variant of this experiment to see how the quality of the review process has changed over time.  This paper was part of the experiment and was therefore assigned to two committees (consisting of reviewers, an Area Chair, and a Senior Area Chair) that reached independent decisions.  If both committees made the same recommendation, this recommendation was followed. If a single committee recommended acceptance, the paper was accepted (with the exception of a few cases in which the other committee identified what we considered a fatal flaw, e.g., an error in a key result).

This copy’s committee reached the following decision: **Accept (Poster)**

The other committee assigned to the paper recommended **Accept (Spotlight)**.  You can find the other set of reviews, along with any follow up discussion with the authors here:
https://openreview.net/forum?id=eIdzV1-Jdwv